# House sparrows do not exhibit a preference for the scent of potential partners with different MHC-I diversity and genetic distances

Luisa Amo[1,2]*, Guillermo Amo de Paz[3], Johanna Kabbert[4,5], Annie Machordom[6]

1 Departamento de Ecología Evolutiva, Museo Nacional de Ciencias Naturales (MNCN-CSIC), Madrid, Spain, 2 Area of Biodiversity and Conservation, Universidad Rey Juan Carlos, Móstoles, Spain, 3 Departamento de Farmacología, Farmacognosia y Botánica, Universidad Complutense de Madrid (UCM), Madrid, Spain, 4 Department of Animal Behaviour, Bielefeld University, Bielefeld, Germany, 5 Department of Experimental Medical Science, Lund University, Lund, Sweden, 6 Departamento de Biodiversidad y Biología Evolutiva, Museo Nacional de Ciencias Naturales (MNCN-CSIC), Madrid, Spain

* luisa.amo@urjc.es

**Data Availability Statement:** All relevant data are within the manuscript and its Supporting information files.

## Abstract

MHC genes play a fundamental role in immune recognition of pathogens and parasites. Therefore, females may increase offspring heterozygosity and genetic diversity by selecting males with genetically compatible or heterozygous MHC. In birds, several studies suggest that MHC genes play a role in mate choice, and recent evidence suggests that olfaction may play a role in the MHC-II discrimination. However, whether olfaction is involved in MHC-I discrimination in birds remains unknown. Previous studies indicate that house sparrow females with low allelic diversity prefer males with higher diversity in MHC-I alleles. Here, we directly explored whether female and male house sparrows (*Passer domesticus*) could estimate by scent MHC-I diversity and/or dissimilarity of potential partners. Our results show that neither females nor males exhibit a preference related to MHC-I diversity or dissimilarity of potential partners, suggesting that MHC-I is not detected through olfaction. Further studies are needed to understand the mechanisms responsible for mate discrimination based on MHC-I in birds.

## Introduction

When selecting partners for mating, females may choose those that will provide the best resources, genes, or both in order to increase offspring fitness [1, 2]. Most experimental studies addressing sexual selection in birds have focused on male quality to explain adaptive mate choice, considering females to assess potential mates by using male ornamental traits, such as plumage coloration [3, 4], or acoustic traits such as male songs [5]. These traits are condition-dependent signals of male quality and therefore, may allow females to obtain resources and/or good genes for their offspring. In contrast, studies addressing the mate choice of females in relation to genetic dissimilarity are less well described. However, the increased use

**Funding:** LA was supported by the Ramón y Cajal program from MINECO (RYC-2012-11353). The study was financed by the Volkswagen Foundation (85 994-1). The funders had no role in study design, data collection and analysis, decision to publish, or preparation of the manuscript.

**Competing interests:** The authors have declared that no competing interests exist.

of molecular techniques for assessing paternity and genetic compatibility shows that females may also choose mates based on traits and/or mechanisms other than the "good genes" hypothesis, thereby arguing for additional mechanism critical to female choosiness. Females may thus use alternative mechanisms for choosing mates such as extra-pair copulations [6, 7], or post-copulatory choice [8]. Furthermore, molecular techniques have also revealed that males also select mates [9]. For example, red jungle fowl (*Gallus gallus*) males decrease their investment in reproduction, i.e., by allocating less sperm, when copulating with females with similar MHC (Mayor Histocompability Complex (see below)) alleles [9], and Leach's storm-petrel (*Oceanodroma leucorhoa*) males avoid pairing with most common MHC homozygous females [10].

The MHC is a large chromosomal region containing several highly polymorphic genes (MHC class I and II loci) that play a central role in controlling immunological self/non-self recognition [11, 12] and encode cell-surface glycoproteins that mediate antigen presentation to T-lymphocytes. Therefore, MHC genes play a fundamental role in immune recognition of pathogens and parasites. Specifically, MHC-I genes are involved in recognition of intracellular pathogens whereas MHC-II genes are involved in the recognition of extracellular pathogens and parasites. For example, in house sparrows, *Passer domesticus*, specific MHC-I alleles are associated with stronger immune T cell responses to MHC-I presented antigen [13] and increased resistance to malaria parasites [14, 15]. Furthermore, some MHC-I genes have been found to partly explain early survival in this species [16]. Consequently, both pathogen-mediated selection [17] and sexual selection [18–21] may explain the maintenance of MHC polymorphism.

The study of sexual selection is providing examples of MHC dissimilar based selection in birds [7, 22, 23], but see [24], here indicating that clutch sizes and the probability to extra-pair relationships can be affected by the partner's MHC diversity [7, 22, 23]. For example, although blue tit (*Cyanistes caeruleus*) females are known to use song [25] and plumage coloration [26] when selecting social mates, there is evidence that extra-pair partners are selected based on their genetic dissimilarity, as extra-pair nestlings showed a higher degree of heterozygosity compared to within-pair nestlings [27]. Females may increase their offspring's fitness by selecting genetically dissimilar mates when engaging in extra-pair copulations as nestlings with a higher degree of heterozygosity were more likely to survive [27]. Although, females may ensure all nestling survival by choosing the best parental male based on phenotypical characteristics such as plumage coloration [26], the use of other characteristics signalling male genetic heterozygosity likely contributes to mate choice in the context of sexual selection [28, 29].

In order to provide the best genetic benefits in relation to the immune systems to their offspring females can select males with "good genes", that will consequently be inherited by their offspring increasing their fitness [30] or select more heterozygous or dissimilar males to increase the genetic diversity of offspring [31]. When this heterozygosity is exhibited in genes of the immune system, the benefits for offspring fitness are obvious in terms of resistance to a wide range of infections [32]. Although, the evidence is not as well-described as in mammals [33, 34], including humans [35], evidence for MHC-associated mate choice has been found in fishes [36] and reptiles [37]. In birds MHC-associated mate choice have been found both for MHC class I [38, 39] and class II [10, 22, 40], but see ([24]). While olfaction has been shown to account for MHC recognition in other taxa, in avian studies sexual selection based on olfaction had been largely neglected in the past. However more recently, two bodies of evidence support olfactory mediated MHC-based mate choice in birds. Firstly, there is an increasing number of studies showing that the chemical profile of bird odour is correlated with MHC-II allelic diversity [41–43]. Secondly, behavioural experiments testing the olfactory preferences of birds have found that olfactory preferences for other conspecifics is based on MHC-II genes [28, 29]. In

support of this, the study by Leclaire and collaborators [28] showed first evidence that birds can use olfactory cues to assess MHC-II similarity. Here, incubating blue petrel, *Halobaena caerulea*, males preferred the scent of MHC-II dissimilar females, while incubating females preferred the scent of MHC-II similar males [28]. Conversely, Grieve and collaborators [29] performed a study assessing the scent preferences in song sparrows, *Melospiza melodia*, and showed that during breeding season both females and males spent longer time periods close to odours from MHC-II dissimilar partners [28]. Given these opposite results found in female preferences of both species, further studies are needed to disentangle the MHC scent preferences of birds in the context of sexual selection during the mating period. Furthermore, to our knowledge, MHC-I scent preferences of birds in the context of sexual selection has not been previously assessed.

In this study we have experimentally explored whether olfactory signals play a role in the preference for potential partners with greater MHC-I dissimilarity and/or diversity in house sparrows, *Passer domesticus*, during the mating period.

## Materials and methods

### Study species

For this study the house sparrow (*Passer domesticus)* was chosen as a model species as their olfactory capabilities have been previously described in the context of roosting cavity assessment [44] and social interactions [45]. Moreover, in the house sparrow, MHC-I allele diversity is associated with reproductive success [46]. It has been shown that males with low MHC-I allelic diversity or too dissimilar MHC-I alleles fail to form breeding pairs [47]. Therefore, female house sparrows likely assess their male mate preferences based on their own MHC-I allelic diversity [39]. Females with low MHC-I allelic diversity choose males with relatively higher MHC-I allelic diversity, whereas no preference is exhibited by females with high or intermediate MHC-I diversity ([39] but see [38]).

We used mist nets to capture 100 male and 51 female adult house sparrows at the Madrid Zoo (Spain) during February and March 2015. Immediately after capturing, all birds were measured with a dial calliper to the nearest 0.01 cm and weighed with a spring balance to the nearest 0.1 g. We measured the length of the visible badge of males with a digital calliper to the nearest mm from the base of the bill to the point on the black breast at which the black feathers finished. This measurement was used instead of the area of the badge, because measurement of the width across the whole black area is less accurate, and it has been used as a measure of badge size in previous studies [48]. All birds were individually banded with numbered aluminium and PVC rings. We obtained blood (80–100 μl) from the brachial vein with the aid of a needle and a capillary tube. Blood samples were centrifuged (2000 × *g*, 5 min) with a portable centrifuge (Nahita 2507–01). Serum and cellular fractions were separated, and the cellular fraction was conserved in 94% ethanol and stored for later analysis.

Birds were housed at the FIEB, Foundation for the Research and the Study of Ethology and Biodiversity (Casarrubios del Monte, Toledo), in outdoor aviaries, until two weeks before the experiments, when birds were individually housed in cages inside the aviaries, so they were maintained at outdoor temperature and photoperiod. Bird weight and badge size was again measured at this time point.

### MHC characterization

For DNA metabarcoding library preparation, PCRs were carried out in a final volume of 25 μl, containing 2.5 μl of template DNA, 0.5 μM of the primers (GCA21M, 5′ GCG T AC A GC G GC T TG T TG G CT G TG A, and fA23M, 5′ GCG CTC CAG CTC CTT CTG CCC

ATA [47], to which the Illumina sequencing adaptor sequences were attached to their 5' ends to allow attaching fragments to the cell flow lawn, 12.5 µl of Phusion DNA polymerase mix (Thermo Fisher Scientific), and ultrapure water up to 25 µl. The reaction mixture was incubated as follows: an initial denaturation at 98 ˚C for 30 s, followed by 35 cycles of 98 ˚C for 10 s, 60 ˚C for 20 s, 72 ˚C for 20 s, and a final extension step at 72 ˚C for 10 minutes. The index sequences (unique short sequences to identify each individual) which are required for multiplexing different libraries in the same sequencing pool were attached in a second PCR round with identical conditions and only 5 cycles. The libraries were run on a 1% agarose gel stained with REAL Safe (Durviz) and imaged under UV light. Negative controls that contained no DNA were included to check for contamination during library preparation. Libraries were pooled in equimolar amounts according to band intensity. The pool was purified using the Mag-Bind RXNPure Plus magnetics beads (Omega Biotek), following the instructions provided by the manufacturer. The pool was sequenced in a MiSeq PE300 run (Illumina).

The data were filtered to remove low-quality sequences, any reads representing artefactual MHC alleles or putatively non-functional sequences [49]. Firstly, a 1% bioinformatics filter per sample was applied [50], to keep only potential true alleles. After that, a minimum total sequence abundance/individual was set to 200, according to the frequencies of our reads and previous research [40]. After data processing, we obtained 117 different MHC-I alleles in 151 individuals. The average (± SE) number of alleles per individual was 4.62 ± 0.15 (range: 1–10). We calculated MHC amino acid diversity for each individual as the number of unique amino acid sequences. The amino acids in the particularly polymorphic peptide-binding regions of the MHC molecules determine what antigens can be bound and are therefore crucial for the function of each allele. We calculated functional diversity for each individual coding the amino acid sequences according to the chemical binding properties of the amino acids [16, 28, 40].

## Amino acid and functional distance between birds

Amino acid and functional distances between individual genotypes were used to describe MHC similarity between individuals and were calculated following the approaches described in [15, 28, 40]. To calculate amino acid distances, a maximum-likelihood tree was inferred for all translated MHC sequences, using PhyML (v. 3.0) [51] in the ATGC platform with a Jones-Taylor-Thornton substitution model [52], selected automatically by Smart Model Selection (SMS) [53], following the Bayesian Information Criterion. Because this metric can be sensitive to cut-off thresholds and other methodological decisions [54, 55], in addition to calculating distances for the full data set, we also generated four additional phylogenies, each removing one of the four alleles with the longest branch lengths. We calculated unweighted UniFrac distances for each of the 5 phylogenies, then used the average of all analyses [29].

To calculate functional distances, we obtained the functional MHC alleles from Lukasch et al. [16], that are based on the chemical binding properties of the amino acids in the peptide binding regions (PBRs), described by five physio-chemical descriptor variables (z-descriptors) for each amino acid [56]. We obtained 113 functional MHC-I alleles. The average (± SE) number of functional alleles per individual was 4.48 ± 0.15 (range: 1–10). The functional MHC alleles obtained are shown in S2 Table. The resulting matrix of allele frequencies was used to construct an alternative maximum-likelihood tree with contml in the PHYLIP-package, v. 3.695. This tree represents clusters of functionally-similar MHC sequences rather than clusters of evolutionary-similar MHC sequences. The amino acid and functional trees were used as references from which the amino acid and functional distances between MHC-sequence repertoires were calculated, using unweighted UniFrac analyses (Phyloseq v. 1.22.3 package in R). We also generated seven additional phylogenies, each removing one of the seven functional

alleles with the longest branch lengths and calculated mean unweighted UniFrac distances across all eight functional trees. Amino acid distances were positively correlated with functional distances among the 151 individuals used in this study (Mantel test: $r = 0.87$, $P = 0.01$; calculated using Vegan v. 2.4–6 package in R).

## Behavioural experiment

The experiment was performed indoors during May and June 2015 using an olfactometry chamber (Fig 1). The experimental device was composed of a small central plastic box (15 x 25 x 25 cm), where the experimental bird was placed, and two lateral chambers referred to as choice chambers (Fig 1). Each choice chamber was divided into two sectors with screens. The distal sectors (15 x 25 x 25 cm) of the choice chambers (30 x 25 x 25 cm) contained two little cages (13.4 x 23.5 x 19.8 cm) where the scent donor birds were situated. Both, the doors connecting the central chamber with the choice chambers and the screens creating the sectors, consisted of a dense plastic mesh that allowed air flow but prevented birds from seeing through them. Overall, the device was sealed and only openings at the farthest walls of the choice chambers allowed air flow. The central chamber contained a small 12 V PC fan that extracted the air from the device creating a controlled low-noise airflow (Fig 1). The fan created two constant air flows, each one entering across the openings located at the farthest walls of each choice chamber, passing by the donor birds, and crossing the central chamber, and going outside from the device through the fan. Thus, the focal bird received two separate air flows, carrying the scent of the corresponding donor bird. Donor birds were kept in darkness (opaque chamber) for the entire trial duration and reduced space (scent donor cages), preventing them from moving or calling. Therefore, the experimental bird only perceived the scent of the donor birds without visual or acoustics contact. The experimental room was to the greatest possible extend sealed from exterior noise, enabling the experimenter to perceive any acoustic signals from any of the birds in the device. The experimenter was present during the entire trial period, but not visible/audible to the focal bird. The device and the methodology have been successfully used in social context studies before [57, 58].

In each test, a bird was introduced into the central box and maintained in the dark for 5 min after that the doors were remotely opened with a rope. We noted down the choice chamber that was first approached by each tested focal bird after opening the central chamber. As the device was opaque and the experimental room was in darkness and silence, the experimenter

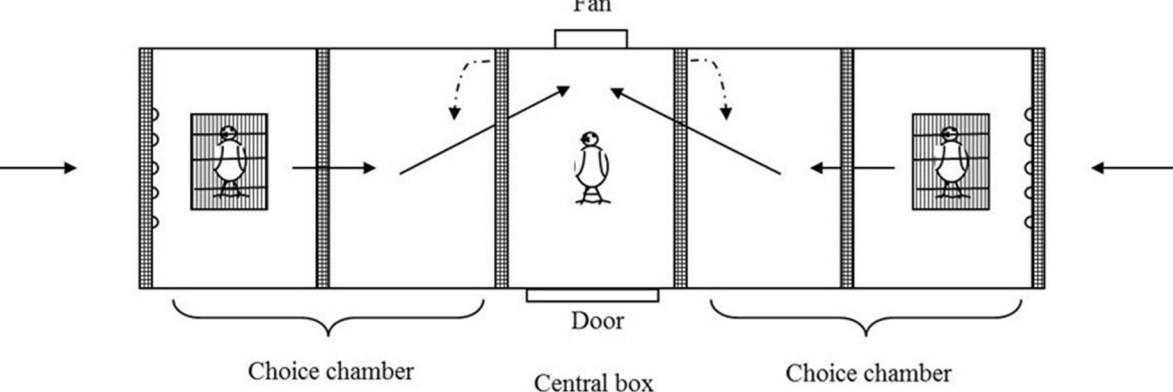

**Fig 1. Olfactometry chamber.** The solid arrows indicate the direction of air flow within the chamber, whereas the dashed lines indicate the direction of opening of the two doors connected to the two choice plastic chambers.

scored by hearing the choice of the focal bird. Immediately after hearing the movement of the bird, the experimenter introduced a hand through the door of the central chamber to ensure the bird was located in the choice chamber where it was heard. The assessment of "first choice" as a preference measure of birds to chemical stimuli has been previously shown [28, 57, 58]. To minimize the experimental time duration of the trials and to release the birds as soon as possible, birds that did not show any choice preference after 1 min, were motivated to make an active choice by gently knocking on the middle of the entry door of the central chamber. The knocking on the door of the central chamber always resulted in the focal bird making a choice. The knocking on the door did not influence the preference of birds (see results). Maximal test time for each bird was 6 minutes.

The two scent donor birds used in the trials were matched for body size, weight, and size of badge in the case of males, to remove any potential correlation between body condition/dominance status and body odour. Thus, any differences in odour profiles between the two scent donor birds could be attributed to MHC and no another confounding factor. Thus, we offered focal birds the scent of two potential mates of similar body condition and dominance status ([59], but see [60]) in the case of males. There were no significant differences in the body condition (repeated measured ANOVA, $F_{1,127} = 0.60$, $P = 0.44$), or the badge size ($F_{1,48} = 0.09$, $P = 0.77$) between both scent donor birds. For methodological problems, we could not perform the MHC-I characterization before the behavioural tests. The MHC-I characterization was done a posteriori and therefore, we could not pair scent donor birds in relation to their MHC-I diversity or dissimilarity to focal birds. We tested 72 males and 48 females. Only one female bird did not move to any chamber even after knocking on the door and was therefore excluded from the analysis. We used 68 different pairs of scent-donor birds (41 pairs of females and 27 pairs of males); each pair was used on average 1.76 (± 0.8) times. Focal birds made 61% of their first choices to the left and 39% of the choices to the right side. There were no significant differences in the number of scent donor birds more diverse or more dissimilar between those located in the right side or in the left side of the olfactory chamber (GLZM, $P > 0.30$ in all cases). In relation to the amino acid diversity, in 47 trials the most diverse bird was located in the right side of the olfactometry chamber and in 50 trials in the left. In relation to the functional diversity, in 49 trials the most diverse bird was in the right side and in 47 trials in the left. In relation to amino acid distance, the most dissimilar bird was in the right side in 56 trials and in the left in 63 trials. In relation to the functional distance, in 54 trials the most dissimilar bird was in the right side and in 65 trials in the left. 68 birds made their choice within less than 1 minute and 33 birds afterwards. Since MHC-I determination was performed a posteriori in the analysis of the preference of focal birds for differences in MHC-I amino acid diversity between scent donor birds, we excluded 22 trials because the difference in the amino acid diversity was 0, which did not allow us to establish a preference. In the analysis of the preference of focal birds for differences in the functional diversity of scent donor birds, we excluded 23 trials because the difference in the functional diversity was 0.

Birds were returned to their cages directly after they were tested. The olfactometry device was carefully cleaned with 94% ethanol in between trials and experiments continued after the ethanol was completely evaporated.

## Data analysis

We performed two sets of analyses to examine scent preferences of focal birds for diversity or similarity of potential partners, always considering the MHC-I of focal birds in the analysis. We included the sex of birds in the analyses to examine whether the sex of the focal bird influenced the preference of birds. We also included the side of the chamber where scent donor

birds where located to control for a potential effect of side in the preference of focal birds. We performed a first set of analyses to examine the preferences of the focal birds for the diversity in amino acid alleles and functional alleles of scent donor birds, considering MHC-I diversity of the focal bird as a covariable. To analyse whether birds could detect amino acid variant numbers of potential mates by using chemical cues alone, we performed a generalized linear mixed model with binomial errors and a logit link function (GLMM). We modelled the probability that birds chose the scent of the conspecific of the opposite sex with greater number of MHC-I amino acid variants (as a dichotomous variable: greater number of MHC amino acid variants (yes) vs. lower number of MHC amino acid variants (not)) in relation to the sex and the number of MHC amino acid variants of the focal bird. We included the interaction between the sex and number of MHC amino acids of the focal bird in the model to test whether males and females differed in their preferences for partners with different numbers of MHC amino acid variants in relation to their own number of MHC amino acid variants. We also included the pair of donor birds in the model as a random factor to control for the fact that pairs of donors could be used more than once. We included in the initial model the side of the chamber where the most diverse scent donor bird was located and a variable reflecting whether the experimental bird left the chamber when we opened the doors or after 1 min as fixed factor.

We performed a similar analysis to assess whether birds chose the scent of the conspecific of the opposite sex with greater number of MHC-I functional alleles, including in number of MHC-I functional alleles of the focal bird in the model, the sex, and the interaction between the number of MHC-I functional alleles of the focal bird and the sex in the model. We also included the pair of donor birds in the model as a random factor. We included in the initial model the side of the chamber where the most diverse scent donor bird was located, and a variable reflecting whether the experimental bird left the chamber when we opened the doors or after 1 min as fixed factor.

In a second set of analyses, we analysed the preference of the focal bird for the MHC-I similarity of scent donor birds, by using the genetic and functional distances between the scent donor birds and the focal bird. We analysed the preference of the focal bird for the MHC similarity of scent donor birds, considering the similarity between the focal bird and the scent donor birds. We performed two generalized linear mixed models with binomial errors and a logit link function (GLMM) to model the probability that birds chose the scent of the conspecific of the opposite sex with greater a) MHC-I amino acid distance and b) MHC-I functional distance (as a dichotomous variable: greater MHC-I amino acid or functional distance (yes) vs. lower amino acid or functional distance (not)) in relation to the sex of the focal bird. We included in the initial model the side of the chamber where the most dissimilar bird was located and variable reflecting whether the experimental bird left the chamber when we opened the doors or after 1 min as fixed factor. We also included the pair of donor birds in the model as a random factor. Data analyses were performed with R program 4.1.2 [61].

In this set of analyses, we considered the choice as a categorical variable, i.e., we analysed whether focal birds chose (or not) the most MHC-I diverse or dissimilar potential partner, without considering the degree of differences between the diversity or similarity of scent donor birds. The analyses and results considering the differences in diversity or dissimilarity of scent donor birds are provided in the supplementary material.

## Ethical note

After capture, birds were housed in ten aviaries (2.5 x 2.5 x 2.5 m), separated by sex (10 females per aviary and 16 males per aviary). Aviaries contained vegetation (bamboo branches) that birds

could use as perches, and grass and sand on the ground. Commercial food for granivorous passerines and water were provided *ad libitum*. To reduce the time of keeping birds separated, two weeks before the experiments, birds were individually housed in cages (60 x 40 x 40 cm) that were located inside the aviaries. Thus, cages were maintained at outdoor temperature and photoperiod. After the behavioural tests were completed, birds were released again in the aviaries for two weeks and thereafter released at their capture site. Birds maintained healthy throughout the experiments. Experiments were carried out under license of the Ethical Committee of the CSIC (268/2015), the Animal Experimental Committee of the Junta de Castilla la Mancha (202121) and the department of Flora and Fauna of Comunidad de Madrid (10/238509.9/14).

## Results

Our results do not show any preference in house sparrows for the scent of potential partners having a greater number of MHC-I amino acid alleles ($Z = 1.08$, $P = 0.06$, Table 1, Fig 2a) or MHC-I functional alleles ($Z = 1.82$, $P = 0.07$, Table 2, Fig 2b). There was no effect of sex in this lack of preferences ($Z = -1.50$, $P = 0.34$, and $Z = -0.17$, $P = 0.86$, respectively), and the interaction between sex and the number of amino acid alleles or functional alleles of the focal bird was not significant in either analysis ($Z = 0.26$, $P = 0.39$ and $Z = -0.39$, $P = 0.70$, respectively). Neither the number of MHC-I amino acid alleles ($Z = -0.31$, $P = 0.14$) nor the number of functional alleles ($Z = -1.53$, $P = 0.13$) influenced the choice of the focal bird. Whether the experimental bird left the chamber when opened or after 1 min did not affect the choice of the respective bird, neither did the side of the chamber where the most diverse scent donor bird was located (see Tables 1 and 2).

Our results do not show any preference of birds for the scent of potential partners having a greater MHC-I amino acid distance ($Z = 0.78$, $P = 0.44$, Table 3, Fig 3a) or functional distance ($Z = 1.61$, $P = 0.11$, Table 4, Fig 3b). Neither the sex ($Z = -0.11$, $P = 0.79$, and $Z = -1.17$, $P = 0.24$, respectively) nor whether the bird made the choice within the first minute or afterwards ($Z = 0.005$, $P = 0.98$, and $Z = 0.68$, $P = 0.50$, respectively) influenced the choice. However, we found a general side bias impacting the analysis of the bird's choice with more birds choosing the right side of the chamber ($Z = -0.83$, $P = 0.03$, and $Z = -2.23$, $P = 0.03$, respectively).

## Discussion

Our experimentally obtained results on house sparrows do not provide evidence of MHC-I scent discrimination in a passerine bird species. Although olfaction has been shown to play a major role in assessing genetic dissimilarity in birds [28, 29] as well as in other taxa [62], our results show that neither females nor males exhibited any preference for the scent of conspecifics with greater MHC-I diversity or dissimilarity. This is contrary to previous studies showing

**Table 1. Model for bird choice in relation to: The number of MHC- I amino acid alleles of a potential partner (as a dichotomous variable: Greater MHC amino acid diversity (yes) vs. lower number of MHC amino acid diversity).**

|  | Estimate | SE | Z | P |
|---|---|---|---|---|
| Intercept | 2.07 | 1.08 | 1.91 | 0.06 |
| Sex | -1.50 | 1.57 | -0.96 | 0.34 |
| Choice within 1 min or after | 0.08 | 0.32 | 0.26 | 0.80 |
| Side | -1.13 | 0.58 | -1.96 | 0.05 |
| MHC amino acid diversity | -0.31 | 0.21 | -1.47 | 0.14 |
| Sex x MHC amino acid diversity | 0.25 | 0.30 | 0.85 | 0.39 |

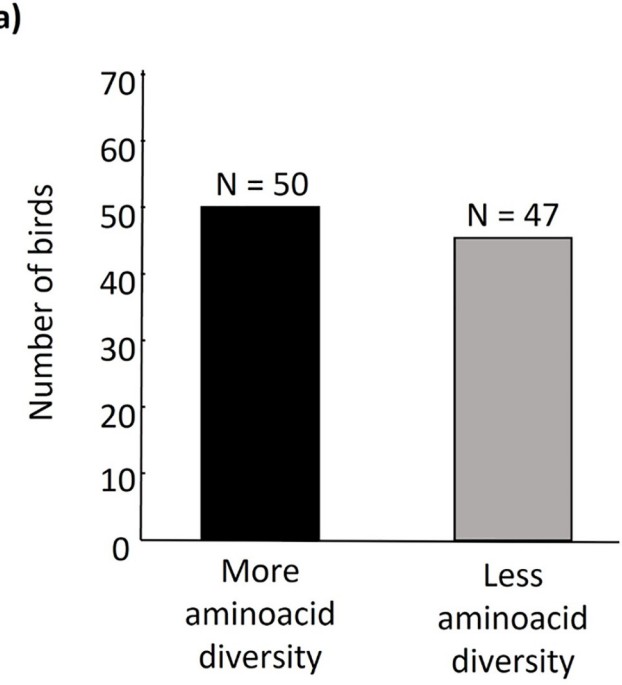

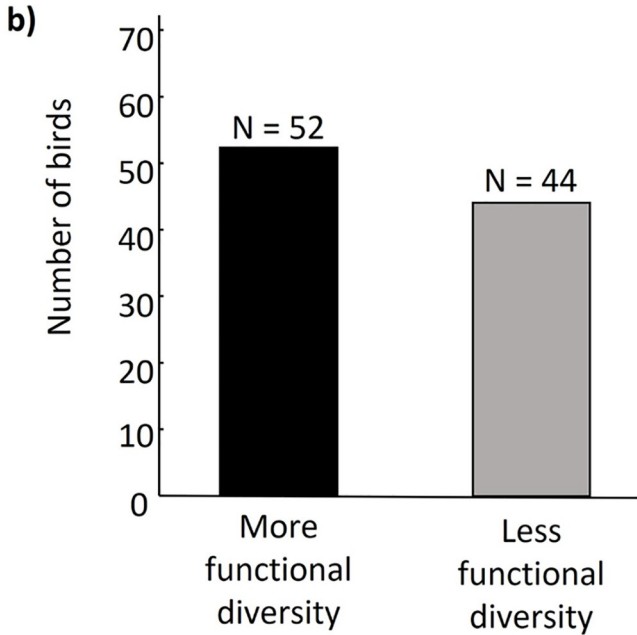

**Fig 2. Number of birds that chose the side of the olfactometry chamber containing the scent of a conspecific of the opposite sex with greater (black) or lower (grey) a) MHC-I amino acid diversity or b) MHC-I functional diversity.**

**Table 2. Model for bird choice in relation to: The number of MHC-I functional alleles of a potential partner (as a dichotomous variable: Greater functional diversity (yes) vs. lower functional diversity).**

|  | Estimate | SE | Z | P |
|---|---|---|---|---|
| Intercept | 1.82 | 1.00 | 1.82 | 0.07 |
| Sex | -0.26 | 1.50 | -0.17 | 0.86 |
| Choice within 1 min or after | 0.15 | 0.28 | 0.54 | 0.59 |
| Side | -0.82 | 0.51 | -1.61 | 0.11 |
| MHC functional diversity | -0.31 | 0.20 | -1.53 | 0.13 |
| Sex x MHC functional diversity | 0.11 | 0.29 | 0.39 | 0.70 |

that female blue petrels (*Halobaena caerulea)* preferred the scent of MHC-II-similar males [28], and female song sparrow (*Melospiza melodia*) preferred the scent of MHC-II-dissimilar and more MHC-II-diverse males during the mating period [29]. Differences between our results using house sparrows and those using blue petrels could be due to different breeding conditions of birds used in the studies. While Leclaire and collaborators tested blue petrels that were already incubating [28] and thus not searching for a partner, we, in contrast, performed our experiment during the breeding period using birds that were separated by sex, preventing pairing or incubating. So, we expect birds would be interested in searching for a partner. Differences between our results and those obtained with song sparrows [29] cannot be attributed to differences in the breeding condition of birds, as in both studies birds were tested during the breeding period and were not incubating. Several explanations may account for the lack of preference for MHC-I dissimilar mates found in our study.

Previous experience with scent donor birds may have masked a preference for MHC-I diverse or dissimilar partners. However, we captured birds during February and March and kept females separated from males. Birds were separated by sex in different aviaries avoiding the establishment of any interaction with potential partners that may have affected olfactory preferences. Therefore, neither the timing of the experiment nor previous interactions during the breeding period or the physiological condition of the birds may explain the lack of preference for MHC-I dissimilar potential partners.

Another possibility to explain the lack of preference observed in our study is that a methodological artifact may have masked preference. In the study with song sparrows, Grieves and collaborators analysed the percentage of time spent close to the two stimuli after 5 minutes from the beginning of the trial. However, neither the first choice nor the preference of birds during the first 5 minutes of the trial was reported. We in contrast, analysed the first choice of birds after 5 minutes of exposure. The validity of first choice as a measure of bird interest in particular chemical stimuli has been previously shown [57, 58, 63–65], including MHC-related scents [28]. In general, first choice is a good measure of the spontaneous interest of an animal to a particular cue, whereas time spent close to a stimulus [66] may rather be related to the

**Table 3. Model for bird choice in relation to: The difference in the MHC-I amino acid distance between the focal birds and the two scent donor birds of the opposite sex (as a dichotomous variable: Greater MHC-I amino acid distance (yes) vs. lower MHC-I amino acid distance (no)).**

|  | Estimate | SE | Z | P |
|---|---|---|---|---|
| Intercept | 0.25 | 0.32 | 0.78 | 0.44 |
| Sex | -0.11 | 0.39 | -0.27 | 0.79 |
| Choice within 1 minute or after | 0.005 | 0.21 | 0.02 | 0.98 |
| Side | -0.83 | 0.38 | -2.19 | 0.03 |

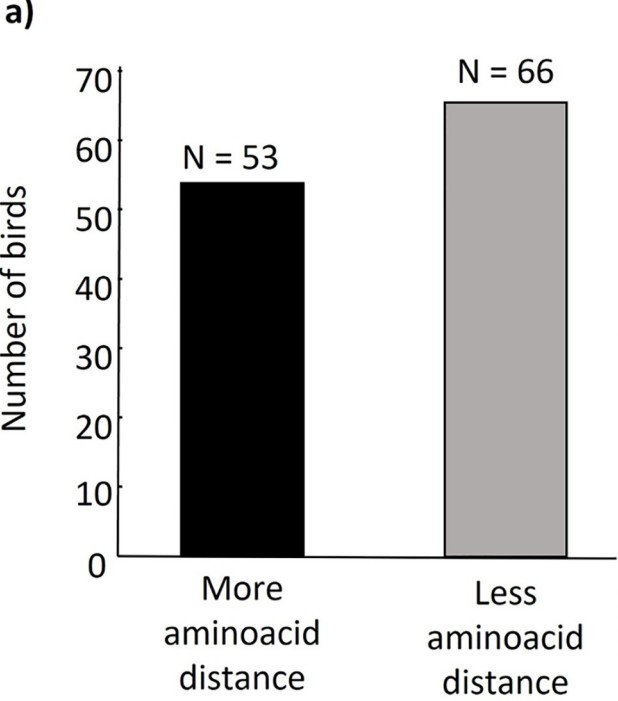

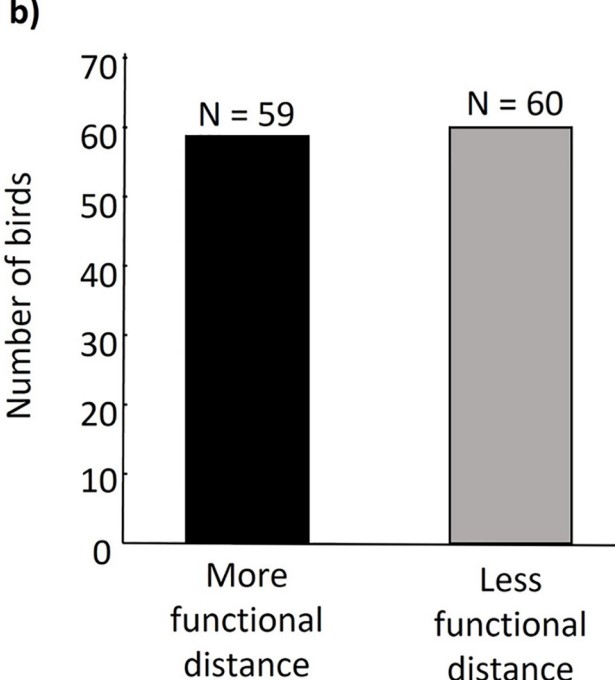

**Fig 3. Number of birds that chose the side of the olfactometry chamber containing the scent of a conspecific of the opposite sex with greater (black) or lower (grey) a) MHC-I amino acid distance or b) MHC-I functional distance.**

**Table 4. Model for bird choice in relation to: The difference in the MHC-I functional distance between the focal birds and the two scent donor birds of the opposite sex (as a dichotomous variable: Greater MHC-I functional distance (yes) vs. lower MHC-I functional distance (no)).**

|  | Estimate | SE | Z | P |
|---|---|---|---|---|
| Intercept | 0.57 | 0.35 | 1.61 | 0.11 |
| Sex | -0.46 | 0.39 | -1.17 | 0.24 |
| Choice within 1 minute or after | 0.15 | 0.22 | 0.68 | 0.50 |
| Side | -0.86 | 0.39 | -2.23 | 0.03 |

behaviour shown later on in the series of events to a certain cue. Using first choice in our experimental set-up using life birds devoid of any other cues, i.e., acoustic ones, reflects a valid measure to assess scent preference. However, to analyse whether the lack of preferences for the scent of most MHC-I diverse or dissimilar conspecifics is maintained over time, further studies are needed to assess the subsequent response of birds to the scent of potential partners.

We found a slight preference for the right side of the chamber that may be attributed to hemispherical asymmetry. House sparrows exhibited laterality in the brain towards the right, as previously reported in other sparrow species, such as American tree sparrows (*Spizella arborea*) [67]. Our results suggest that this laterality does not seem to affect olfactory preferences of focal birds. However, further studies are needed to explore olfactory laterality in avian species.

In addition, the lack of odour-based preference found in our study for more diverse or more dissimilar potential partners is unlikely the result of low olfactory capacities because house sparrows are known to use olfaction in social contexts [45].

Nevertheless, we consider our results robust due to the large sample size and the elimination of potential confounding factors i.e., body condition bias by using scent-donor birds of similar body condition. Based on preliminary statistical analysis using the difference in MHC-I diversity or dissimilarity between scent donor partners we observed that these differences did not explain the choice of birds (see S1 Table). Therefore, we can exclude that differences in MHC-I diversity or similarity were very small and house sparrows were not able to detect them. Moreover, due to expected differences in the volatile profile of feathers and uropygial gland secretions [68] we used live birds as scent sources as opposed to merely uropygial gland secretions to increase the robustness of our study approach.

However, while there is no reason to expect that all bird species use the same type of information to evaluate potential partners, our results are difficult to explain in a sexual context, because in house sparrows MHC-I diversity is associated with reproductive success [47]. Accordingly, house sparrow males with a low degree of MHC-I diversity or too dissimilar MHC-I alleles failed to form breeding pairs [47] and house sparrow females with low MHC-I allelic diversity choose males with high MHC-I allelic diversity [39], suggesting that house sparrow females show mate preferences based on the MHC-I status of potential partners ([39], but see [38]).

Collectively, the contradictory results of our study and others suggest that more research is needed to examine the mechanisms in more detail that may account for the olfactory assessment of MHC-I characteristics in birds. Although, several hypotheses have been proposed to explain the underlying mechanisms [62], it is still not completely understood how MHC genes influence scent [69, 70]. As MHC proteins appear in urine and sweat, it was proposed that these molecules may constitute odorants [71]. However, since MHC proteins are non-volatiles, this hypothesis is rather weak [72]. While MHC-related peptides may potentially be detected by the vomeronasal organ [73] in mammals, in birds, that lack a vomeronasal organ, this is less

likely. An alternative hypothesis suggests that the peptides bound by MHC molecules are the precursors of volatile odorants [74], but this remains to be tested in birds. It has further been proposed that MHC genes may influence the bacterial profile that influences the scent of an individual [75–78] and that MHC-related scent could also be produced by symbiotic bacteria [79], whose diversity is associated with MHC-II diversity [77]. In many avian species, the main source of scent that birds can detect usually comes from externally applied secretions of the uropygial gland [80]. The quantity and composition of this secretion varies between species, sexes, ages, seasons, and is dependent on diet and hormonal levels [81]. This secretion conveys information on genetic compatibility [28, 42], which may play a role during kin recognition [82] and mate choice [28, 29, 40]. Evidence that birds are using the volatile compound of uropygial gland secretions for detecting the MHC-II characteristics of conspecifics [29] comes from a study in song sparrows that were able to assess MHC-II similarity when exposed only to the uropygial gland secretion. While a recent body of evidence suggests that MHC-II genes play a critical role in mate choice that can be assessed by olfaction as shown for blue petrels [28] and song sparrows [29], whether house sparrows employ olfaction to assess MHC-II genes or other genetic characteristics of potential partners is still not known.

Similarly, further research is needed to elucidate the source of MHC-I-related scent cues in birds. One of the major outstanding questions is whether house sparrow body odour does indeed convey information about the genetic make-up of MHC-I. We therefore propose further experiments examining whether chemical profiles vary in relation to MHC-I characteristics of individuals and to determine whether these potential differences in MHC-I characteristics can be assessed through olfaction in birds.

In conclusion, the observed lack of preference for the scent of MHC-I dissimilar potential partners suggests that the assessment of genetic diversity and dissimilarity in MHC-I is not based on olfaction in our study population of house sparrows. However, our results do not fully exclude the possibility that mate choice in house sparrows includes the selection of partners based on the degree of similarity or dissimilarity in MHC-I genes using other mechanisms rather than olfaction. Here, effects of increased MHC-I allele diversity in the progeny may result in higher resistance to pathogens [13] such as malaria parasites [14, 15, 83], that may increase reproductive success [47]. We suggest that different methodological approaches in future investigations may further the understanding of the mechanisms responsible for MHC-I based mate discrimination in birds.

## Supporting information

**S1 Table. Data obtained in the behavioural experiment.**
(DOCX)

**S2 Table. Aminoacid and functional diversity and distance of experimental birds.**
(DOCX)

**S3 Table. Analysis of the choice of birds in relation to differences in the diversity or dissimilarity between scent donor birds.**
(DOCX)

## Acknowledgments

We especially thank the associate editor, Dr. Rodríguez Ruiz, and the anonymous referees for their useful comments. We also thank Pilar Ochoa for laboratory analysis, Agustín López Goya, and the personnel of Madrid Zoo for allowing the capture of birds in the Zoo.

## Author Contributions

**Conceptualization:** Luisa Amo.

**Formal analysis:** Luisa Amo, Guillermo Amo de Paz, Annie Machordom.

**Funding acquisition:** Luisa Amo.

**Investigation:** Luisa Amo, Johanna Kabbert.

**Methodology:** Luisa Amo, Johanna Kabbert.

**Writing – original draft:** Luisa Amo, Guillermo Amo de Paz, Johanna Kabbert, Annie Machordom.

**Writing – review & editing:** Luisa Amo, Guillermo Amo de Paz, Johanna Kabbert, Annie Machordom.

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
