## [Decision Letter · Decision Letter 0]

28 Mar 2022

PONE-D-22-00952House sparrows do not exhibit a preference for the scent of potential partners with different MHC-I allele numbers and genetic distancesPLOS ONE

Dear Dr. Amo,

Thank you for submitting your manuscript to PLOS ONE. After careful consideration, we feel that it has merit but does not fully meet PLOS ONE’s publication criteria as it currently stands. Therefore, we invite you to submit a revised version of the manuscript that addresses the points raised during the review process.

Both reviewers and myself agree in that this is a very interesting paper, although it would be necessary to perform some changes before accept it for publication. Please submit your revised manuscript by May 07 2022 11:59PM. If you will need more time than this to complete your revisions, please reply to this message or contact the journal office at plosone@plos.org. Please include the following items when submitting your revised manuscript:A rebuttal letter that responds to each point raised by the academic editor and reviewer(s). You should upload this letter as a separate file labeled 'Response to Reviewers'.A marked-up copy of your manuscript that highlights changes made to the original version. You should upload this as a separate file labeled 'Revised Manuscript with Track Changes'.An unmarked version of your revised paper without tracked changes. You should upload this as a separate file labeled 'Manuscript'.

We look forward to receiving your revised manuscript.

Kind regards,

Magdalena Ruiz-Rodriguez

Academic Editor

PLOS ONE

Journal Requirements:

2. In your Methods section, please include a comment about the state of the animals following this research. Were they euthanized or housed for use in further research? If any animals were sacrificed by the authors, please include the method of euthanasia and describe any efforts that were undertaken to reduce animal suffering.

(LA was supported by the Ramón y Cajal program. The study was funded by the Volkswagen Foundation (85 994-1).)

Additional Editor Comments (if provided):

The manuscript is well written and easy to follow, and the experimental design as well as the results analyses are correct.

Maybe it is not very clear in the variables such as "amino acid variants", "amino acid distances" and "functional alleles", the mening of "more" and "less", when it is considered in each category.

Reviewers' comments:

Reviewer's Responses to Questions

**Comments to the Author**

1. Is the manuscript technically sound, and do the data support the conclusions?

Reviewer #1: Yes

Reviewer #2: Partly

2. Has the statistical analysis been performed appropriately and rigorously? 

Reviewer #1: Yes

Reviewer #2: Yes

3. Have the authors made all data underlying the findings in their manuscript fully available?

Reviewer #1: Yes

Reviewer #2: No

4. Is the manuscript presented in an intelligible fashion and written in standard English?

Reviewer #1: Yes

Reviewer #2: Yes

5. Review Comments to the Author

Reviewer #1: The study investigates whether assortative mating for MHC-I diversity and dissimilarity is mediated by olfactory cues in a passerine bird, the house sparrow. Specifically, the authors experimentally test through behavioural observations if males and females show a preference for potential mating partners that differ in MHC-I amino acid and functional variance as well as in their similarity. Neither males nor females showed a significant preference for conspecifics differing in MHC-I amino acid and functional diversity or dissimilarity. Avian olfaction has recently gained interest in ornithology and behavioural ecology. However, the implications of olfactory perception in bird biology and evolution are for the most part still obscure. This study tries to fill the knowledge gap. The experimental hypothesis builds on a solid amount of literature and I’m glad that it has finally been tested, especially in this bird species. The manuscript is well-written, with sound statistical methods, and the conclusions follow the results obtained. However, some sections should be further improved before acceptance, in particular the discussion (see below).

Major comments:

1. ll. 53-56: Not clearly written. I guess here you want to say that the increased use of molecular techniques for assessing paternity and genetic compatibility showed that females can choose mates also based on traits and/or mechanisms other than the “good genes” hypothesis? Please, clarify. Also, keep in mind that mate choice can also occur in males. Since this study tested also for male mate choice I would definitely not narrow the reasoning to female mate choice in this entire paragraph. Finally, I would like to see a (review) reference to the different mechanisms of assortative mating to which you are referring to.

2. ll. 56-57. Extra-pair copulations and post-copulatory choice are specific mechanisms through which females modify the paternity of the offspring rather than mechanisms of choice of (social) mates and I would be careful in mixing the two. Please, explain better or remove the sentence.

3. ll. 65-67. From this sentence it is unclear why females should do that only when engaging in extra-pair copulations. Please, add details or remove “when engaging in extrapair copulations”.

4. ll. 124-126. As far as I understand the badge size was used to account for male dominance status (l. 229). I’m not against doing this but the reliability of the badge size as signal of dominance in house sparrows has recently been questioned and the reader should be warned about this. See: Sanchez-Tojar, Alfredo, et al. "Meta-analysis challenges a textbook example of status signalling and demonstrates publication bias." Elife 7 (2018): e37385, https://doi.org/10.7554/eLife.37385.

5. l. 201. How were doors opened? Was it remotely or the experimenter physically got close to the bird and opened them manually? Add details to understand if the focal bird may have been startled during the opening phase.

6. ll. 221-224. How long did every test lasted?

7. l. 226. How did you calculate body condition? Was it just body weight?

8. ll. 234-235. Out of the 68 pairs, how many were males and how many females? Only pairs of the opposite sex respect to the focal bird were tested, correct? State it clearly.

9. ll. 319-321. Before discussing the methodological differences, there are two (quite obvious) hypotheses that are not being mentioned and that I recommend discussing. First, passerine birds are among the bird species with the lowest olfactory bulb size (maybe this could also be mentioned in the introduction?). Hence, a reduced olfactory performance compared to other species, in particular versus procellariiformes that are on the opposite side of the spectrum, can be expected. See for instance: Corfield, Jeremy R., et al. "Diversity in olfactory bulb size in birds reflects allometry, ecology, and phylogeny." Frontiers in neuroanatomy 9 (2015): 102, https://doi.org/10.3389/fnana.2015.00102.

Second, there is no mention to the hypothesis that house sparrows may perceive the odour cues and mate assortatively based on the MHC-II alleles (not investigated here) rather than based on the MHC-I alleles. This is particularly relevant as some studies that you also mention (references nr. 7, 9, 15, 16 and 66) and that are central in supporting the hypothesis on the significant role of MHC in avian mate choice, focused on MHC-II alleles instead of MHC-I alleles.

10. l. 346. Could the use of olfactory cues for mate choice be seasonal?

11. ll. 372-374. This information has already been mentioned earlier in the study and it does not lead to further discussion here. I suggest to remove it.

12. l. 634. Try to merge together table 1 and 3 as well as tables 2 and 4 if possible. The legend of each table can be reduced since all the information is also in the methods section.

Minor comments:

13. l. 26. …selecting males with genetically compatible or heterozygous MHC.

14. l. 27. evidence suggests.

15. l. 31. number of MHC.

16. ll.32-35. The sentence is long and hard to read. Please, split it in two parts.

17. l. 36. responsible for mate discrimination based on MHC-I in birds.

18. ll. 45-50. Too long sentence. Please, split it in two parts. For instance “…[5]. These traits are…”.

19. ll. 50-51. The sentence doesn’t really follow the previous one. Move it after the first sentence of the introduction or remove it.

20. l. 58. Remove “alternative”. Alternative sexual selection doesn’t really exist.

21. l. 60. extra-pair relationships can. Please write “extra-pair” consistently throughout the text.

22. l. 63. Remove “in”.

23. l. 70. characteristics signaling male genetic heterozygosity.

24. l. 80. Write MHC-I (or MHC class I) and MHC-II consistently throughout the text.

25. l. 90. Well-described.

26. l. 93. Parenthesis opens but doesn’t close.

27. l. 99. During the breeding season.

28. l. 100. Females and males.

29. l. 133. 100% ethanol? Add the percentage.

30. l. 151. Remove space before %.

31. l. 199. Low noise-controlled.

32. l. 220. Differently from Mathematics, in Biology it’s close to impossible to “demonstrate” something as new evidence can change our theories and hypotheses. It’s better to prefer the wording “shown” or synonyms.

33. l. 237. Which kind of alcohol? Ethanol? Pure or diluted? Specify the percentage.

34. l. 252. donors were used.

35. l. 273. Missing space between “X” and “2.5”. Use the “X” symbol consistently (see also l. 278).

36. l. 277. birds used in the trials.

37. l. 290. Any bird preference for.

38. l. 299. Remove “as fixed factor” as you already mentioned it in the methods”.

39. l. 313. See comment 32.

40. l. 326. See comment 32.

41. l. 328. Remove the second “used”.

42. l. 348. [53], it is still.

43. l. 375. Investigations.

44. l. 403. Remove space between “extra-“ and “pair”.

Reviewer #2: This paper tests for evidence of odour-based discrimination in birds based on MHC class I diversity and similarity. As an understudied area of research, this paper makes a useful contribution to the literature, but I have two main concerns I would like to see addressed before recommending for acceptance: presentation of methods and data, and the focus and scope of the discussion. I discuss in more detail below.

Presentation of methods and data:

It is essential to present at least a summary (e.g. mean +/- SE) of the MHC class I diversity and distances of the stimuli presented to the focal birds. Such data are necessary to evaluate the differences in allelic diversity and genetic similarity of the paired stimuli birds were tested with. For example, it is possible that birds did not respond differentially to donor bird odours because differences in MHC diversity +/or similarity were very small – if that may be the case, it is important to discuss whether a detection capability threshold based on MHC diversity/similarity was likely to have been met or not.

Relatedly, if the distances and diversity differences were very small, I recommend the authors analyze the subset of maximally different stimuli pairings (using some reasonable threshold determined a priori) to see if the focal birds presented with maximally different pairings responded differently.

Given that these were post hoc tests, it seems that these behavioural trials were not originally designed to test MHC-based odour preferences. The authors should be clear about this and state how and when these data were originally used. I don’t necessarily think it’s inappropriate to evaluate the data post hoc, but I think it is important to be clear and up front about this, and to clearly reference any other papers in which these data have been previously used.

Focus and scope of discussion:

Related to the methods and study design, the authors should discuss limitations of the study. For example, with respect to whether the MHC diversity and similarity distances were likely appropriate/sufficient for detection by house sparrows.

Currently, the authors spend a lot of time discussing the validity of using first choice, and of tapping on the maze after 1 min to force a choice, but they do not discuss the possibility that this method may not be as appropriate for passerines or sparrows in particular, compared to other species in which first choice is more commonly used, such as seabirds, which may reasonably be expected to behave differently in the test chamber than sparrows do. The authors argue that many studies use this method, but they do not point out that at least as many studies (if not more) use cumulative time spent with a sample type, also with good success (especially for passerines).

Finally, there is no discussion of how the timing of the experiment and the physiological condition of the birds could have impacted results or predictions of focal bird preference. In the Leclaire study referenced, birds were incubating and may have preferences related to this, whereas in the Grieves study birds were in breeding condition and may have been motivated differently than in the Leclaire study. This should be included in the discussion alongside consideration of the breeding condition of the birds used in the author’s present study and how this may have affected results and/or predictions/expected outcomes.

Detailed comments:

Line 86: why would MHC I be related to bacterial loads?

Lines 93-101: odd to name authors of one study but not the other, and name the study species in one study but not the other. Suggest rephrasing both sentences for consistency of results presentation. It is also worth explaining here or in the discussion the different predictions that can reasonably be made for each sex at different breeding stages (such as during mate choice and pairing vs during incubation) or different predictions that might be made for different taxonomic groups (seabirds vs songbirds) that may differ in level of parental care and investment between the sexes (sex roles, timing of care/investment for each sex, etc).

Line 101: results of Leclaire and Grieves are conflicting but are they controversial? Perhaps not, especially when considering time in breeding cycle as described above.

Line 115: MHC I, II, or both?

Line 120: high or intermediate MHC diversity

Line 133: when was the genotyping performed? Birds were captured Feb-Mar 2015 but it’s not clear when genotyping was done in relation to when the behavioural trials were conducted.

Line 136: given that all birds were initially allowed to interact in a group aviary, could it be that mate choice had already taken place, or preferences have otherwise been established? Should discuss whether/how this may have impacted results. Was any evidence of courtship or breeding activity observed during the time birds were group housed?

Line 143: include primer references/original source for primer development

Line 144: please describe what is meant by 'illumina sequencing primer sequences' in more detail so methods can be evaluated.

Line 148: please explain how index sequences differ from illumina sequencing primer sequences and why these were attached in a separate round of PCR

Line 149: demultiplexing?

Line 158: what criteria were used to set this 1% threshold? Overall, more information on sequence data processing methods is needed to properly evaluate these methods. The only reference given is 10 years old and, given the rapidly changing nature of data handling methods in this field, it is essential to ensure the most up to date and appropriate methods are being used for this type of data.

Line 159-160: rarefaction is not appropriate for these data (see McMurdie, P.J. and Holmes, S., 2014. Waste not, want not: why rarefying microbiome data is inadmissible. PLoS computational biology, 10(4), p.e1003531.).

In Strandh et al. 2012, this 200 cutoff was based on inclusion of technical duplicates, which his reliable only within the sequencing run being evaluated. Were technical duplicates included in this study? If so, this procedure must be described and explained. If not, this method is inappropriate and should not be used. Instead, please update with more recent/current methods.

Line 169-170: JTT SMS explain/write out in full on first mention

Line 201: was the light turned on or was it kept dark throughout the trial?

Line 217: was experimenter present the entire time? Was experimenter visible to the focal / test birds? Was the test chamber opaque or transparent?

Line 224: admittedly, I struggle to accept this method as being appropriate... Regardless, it is important to present a summary of how many birds was this done for. A 1 min trial duration seems very short. Many similar studies allow 15 min for birds to choose, even those scoring first choice. I recognize you cannot change this now, but in future, why not use a longer trial duration and evaluate both first choice and choice / time?

Line 232: why were only 120 of the original 151 birds tested?

Line 235: please include discussion of the side bias exhibited by test birds. How often was the more vs less diverse and more vs less similar individual placed in the left or right side of the olfactometry chamber?

Line 237: what type and % of alcohol was used?

Line 266: what were the distances? How was it determined that this was a detectable threshold or that differences in allelic diversity and genetic similarity were otherwise reasonable? Must present data on what MHC distance and allelic diversity pairings were used.

Line 286: these trials appear to have been used for multiple experiments/to test multiple questions post hoc. If so, this should be clearly stated somewhere in the methods.

Line 316-321: again it's odd presentation to name author in one study and not the other, suggest rephrase so both are consistent.

Line 321: song sparrows are relatively asocial, blue petrels relatively social, and house sparrows relatively social, but all species may differ in level of sociality during breeding – please discuss how you think sociality may be involved.

Line 326: many of these studies set a maximum trial duration of 15 min, presumably to allow time for the birds to choose on their own. The present study seems to allow the birds only 1 min, followed by tapping on the test chamber to force the birds to move, which while use in prior studies of this group. The potential impacts of this should be discussed.

Line 327: but what MHC differences were used in terms of both diversity and dissimilarity? Again, this information needs to be provided.

Line 331-341: this would be a good place to discuss the results in terms of the relative differences in MHC I diversity and similarity of the stimuli, and how this may have affected the results.

Line 342: returning back to a justification of the first choice method is jarring here; what does this have to do with future work or using preen oil instead of whole body odour? Why would preen oil be superior to use of whole body odour? Suggest restructuring the relevant paragraphs for clarity/flow.

Line 347: potential mechanisms for scent-cues of MHC genotype would be better placed in the introduction to set up the study rationale. Since the study did not find evidence for odour choice, I don’t think this discussion of potential mechanisms needs to be here.

Line 353: birds lack a VNO – what is the relevance of including this information here?

Line 354: would MHC I be expected to be involved in influencing bacterial profiles?

Lines 357-360: please put this information in the context of the present study.

Line 374: a lot of time is spent early in the discussion justifying the use of first choice, but now seems to conclude by saying that different methods should be used - what methods should be used/are recommended? Isn't it worth suggesting that choice over time (cumulative time spent with stimulus) might be a good approach to consider for this type of research in songbirds? First choice has been used successfully in several studies, but so has cumulative time spent, and this should be acknowledged and evaluated. Overall, I think the discussion should be reorganized and more tightly focused around the research question and study results.

6. PLOS authors have the option to publish the peer review history of their article (what does this mean?). If published, this will include your full peer review and any attached files.

Reviewer #1: No

Reviewer #2: No

---

## [Author Response · Author response to Decision Letter 0]

30 May 2022

Response to comment of Reviewer 1

Review Comments to the Author

Reviewer #1: The study investigates whether assortative mating for MHC-I diversity and dissimilarity is mediated by olfactory cues in a passerine bird, the house sparrow. Specifically, the authors experimentally test through behavioural observations if males and females show a preference for potential mating partners that differ in MHC-I amino acid and functional variance as well as in their similarity. Neither males nor females showed a significant preference for conspecifics differing in MHC-I amino acid and functional diversity or dissimilarity. Avian olfaction has recently gained interest in ornithology and behavioural ecology. However, the implications of olfactory perception in bird biology and evolution are for the most part still obscure. This study tries to fill the knowledge gap. The experimental hypothesis builds on a solid amount of literature and I’m glad that it has finally been tested, especially in this bird species. The manuscript is well-written, with sound statistical methods, and the conclusions follow the results obtained. However, some sections should be further improved before acceptance, in particular the discussion (see below).

Response: Thanks, we have tried to improve the ms following your suggestions.

Major comments:

1. ll. 53-56: Not clearly written. I guess here you want to say that the increased use of molecular techniques for assessing paternity and genetic compatibility showed that females can choose mates also based on traits and/or mechanisms other than the “good genes” hypothesis? Please, clarify. Also, keep in mind that mate choice can also occur in males. Since this study tested also for male mate choice I would definitely not narrow the reasoning to female mate choice in this entire paragraph. Finally, I would like to see a (review) reference to the different mechanisms of assortative mating to which you are referring to.

Response: Thanks, we have tried to improve this paragraph (lines 59-60): “In contrast, studies addressing the mate choice of females in relation to genetic dissimilarity are less well described. However, the increased use of molecular techniques for assessing paternity and genetic compatibility showed that females can choose mates also based on traits and/or mechanisms other than the “good genes” hypothesis, but indeed suggest additional mechanism critical to female choosiness. Females may thus use alternative mechanisms for choosing mates such as extra-pair copulations [6, 7], or post-copulatory choice [8]. Furthermore, molecular techniques have also revealed that male choice may also use alternative mechanisms for selecting females (Gillingham et al. 2009). For example, red jungle fowl (Gallus gallus) males decrease their investment in reproduction, i.e., by allocating less sperm, when copulating with females with similar MHC alleles (Gillingham et al. 2009).” 

2. ll. 56-57. Extra-pair copulations and post-copulatory choice are specific mechanisms through which females modify the paternity of the offspring rather than mechanisms of choice of (social) mates and I would be careful in mixing the two. Please, explain better or remove the sentence.

Response: Our aim was to describe mate selection, not only social partner selection, so, we have removed the sentence: “In fact, in many passerine species, male parental care of offspring is needed to ensure nestling survival.” 

3. ll. 65-67. From this sentence it is unclear why females should do that only when engaging in extra-pair copulations. Please, add details or remove “when engaging in extrapair copulations”.

Response: Thanks, we have tried to explain this better “For example, although blue tit Cyanistes caeruleus females are known to use song [12] and plumage coloration [13] when selecting social mates, there is evidence that they select extra-pair partners based on their genetic dissimilarity, as extra-pair nestlings showed higher degree of heterozygosity compared to within-pair nestlings [14]. Since nestlings with higher degree of heterozygosity were more likely to survive [14], females may increase their fitness by selecting genetically dissimilar mates when engaging in extra-pair copulations. Although, females may ensure all nestling survival by choosing the best parental male based on phenotypical characteristics such as plumage coloration [13], they may indeed use other characteristics signalling male genetic heterozygosity [15, 16].

4. ll. 124-126. As far as I understand the badge size was used to account for male dominance status (l. 229). I’m not against doing this but the reliability of the badge size as signal of dominance in house sparrows has recently been questioned and the reader should be warned about this. See: Sanchez-Tojar, Alfredo, et al. "Meta-analysis challenges a textbook example of status signalling and demonstrates publication bias." Elife 7 (2018): e37385, https://doi.org/10.7554/eLife.37385.

Response: Thanks, we have added this reference.

5. l. 201. How were doors opened? Was it remotely or the experimenter physically got close to the bird and opened them manually? Add details to understand if the focal bird may have been startled during the opening phase.

Response: We have added that “the doors were remotely opened with a rope”. Therefore, the chamber was not directly approached by the experimenter and did not scare the birds while opining the doors.

6. ll. 221-224. How long did every test lasted?

Response: We have explained that “Tests lasted up to 6 minutes”.

7. l. 226. How did you calculate body condition? Was it just body weight?

Response: Yes, we have changed body condition to body weight. 

8. ll. 234-235. Out of the 68 pairs, how many were males and how many females? Only pairs of the opposite sex respect to the focal bird were tested, correct? State it clearly

Response: We have added that we used: “41 pairs of females and 27 pairs of males as scent donours”. We already said that we offered focal birds the scent of two potential mates of similar body condition and dominance status in the case of males. So, it is stated.

9. ll. 319-321. Before discussing the methodological differences, there are two (quite obvious) hypotheses that are not being mentioned and that I recommend discussing. First, passerine birds are among the bird species with the lowest olfactory bulb size (maybe this could also be mentioned in the introduction?). Hence, a reduced olfactory performance compared to other species, in particular versus procellariiformes that are on the opposite side of the spectrum, can be expected. See for instance: Corfield, Jeremy R., et al. "Diversity in olfactory bulb size in birds reflects allometry, ecology, and phylogeny." Frontiers in neuroanatomy 9 (2015): 102, https://doi.org/10.3389/fnana.2015.00102. Second, there is no mention to the hypothesis that house sparrows may perceive the odour cues and mate assortatively based on the MHC-II alleles (not investigated here) rather than based on the MHC-I alleles. This is particularly relevant as some studies that you also mention (references nr. 7, 9, 15, 16 and 66) and that are central in supporting the hypothesis on the significant role of MHC in avian mate choice, focused on MHC-II alleles instead of MHC-I alleles.

Response: In relation to the first hypothesis you mention, there is evidence that house sparrows can use olfaction in social contexts. We have mentioned this in the discussion: “The absence of odour-based preference for more diverse or more dissimilar potential partners could hardly be explained by a lack of olfactory abilities in house sparrows because house sparrows are known to use olfaction in social contexts [Fracasso et al. 2019], so a low olfactory capability is also not likely to explain this lack of preference.” In relation to the second hypothesis you suggest, it is true that house sparrows may perceive the odour cues related to MHC-II alleles instead of MHC-I alleles, but previous studies have shown that they can mate assortatively based on the MHC-I alleles [42], so we think it is worth to examine whether they may also assess the MHC-I allele characteristics of potential partners. Ultimately, we have mentioned that maybe house sparrows use olfaction to assess MHC-II status of potential partners: “However, while there is no reason to expect that all bird species use the same type of information to evaluate potential partners, our results are difficult to explain in a sexual context, because in house sparrows MHC-I diversity is associated with reproductive success [40]. It is also known that males with low degree of MHC-I diversity or too dissimilar MHC-I alleles fail to form breeding pairs [41]. Furthermore, results of a previous study showed that house sparrow females with low MHC-I allelic diversity choose males with high MHC-I allelic diversity [42], suggesting that house sparrow females show mate preferences based on the MHC-I of potential partners ([42], but see [43]). Collectively, in this study we tested whether olfaction is mediating olfactory recognition of MHC-I, but further experiments are needed to test whether house sparrows may use olfaction to assess MHC-II similarity of potential partners, as has been shown in blue petrels [15] and song sparrows [16].”

10. l. 346. Could the use of olfactory cues for mate choice be seasonal?

Response: Yes, the response to olfactory cues of conspecifics may be seasonal, but we performed the experiment during the breeding period with birds that were not reproducing, as they were separated by sex. Therefore, we may expect they were looking for a partner. We have explained in the discussion: “Differences between our results with house sparrows and those with blue petrels can be due to differences in the breeding conditions of birds between the studies, because Leclaire and collaborators tested blue petrels that were incubating [15] and thus, not searching for a partner. In contrast, we performed the experiment during the breeding period with birds that were separated by sex, and therefore, they were not paired or incubating, so, they may be interested in searching for a partner. Differences between our results and those obtained with song sparrows cannot be attributed to differences in the breeding condition of birds, as in both studies birds were tested during the breeding period and were not incubating. Several explanations may explain the lack of preference for MHC-I found in our study system. Previous experience with scent donour birds may have masked a preference for MHC-I diverse or dissimilar partners. However, we captured birds during February and March and kept females separated from males. Birds were separated by sex in different aviaries, to avoid establishing any interaction with potential partners that may have affected olfactory preferences. Therefore, neither the timing of the experiment nor previous interactions during the breeding period or the physiological condition of the birds explain the lack of preference for MHC-I dissimilar potential partners.”

11. ll. 372-374. This information has already been mentioned earlier in the study and it does not lead to further discussion here. I suggest to remove it.

Response: We have removed the sentence.

12. l. 634. Try to merge together table 1 and 3 as well as tables 2 and 4 if possible. The legend of each table can be reduced since all the information is also in the methods section.

Response: Thanks, we have merged table 1 and 3 in table 1a. and 1.b, and table 2 and 4 in table 2.a. and 2.b. and we have reduced the legend. 

Minor comments:

13. l. 26. …selecting males with genetically compatible or heterozygous MHC.

Response: Thanks, we have changed the sentence.

14. l. 27. evidence suggests.

Response: Thanks

15. l. 31. number of MHC.

Response: Thanks

16. ll.32-35. The sentence is long and hard to read. Please, split it in two parts.

Response: thanks, we have split it. 

17. l. 36. responsible for mate discrimination based on MHC-I in birds.

Response: Thanks.

18. ll. 45-50. Too long sentence. Please, split it in two parts. For instance “…[5]. These traits are…”.

Response: Thanks, we have split it.

19. ll. 50-51. The sentence doesn’t really follow the previous one. Move it after the first sentence of the introduction or remove it.

Response: We have removed it.

20. l. 58. Remove “alternative”. Alternative sexual selection doesn’t really exist.

Response: We have removed it.

21. l. 60. extra-pair relationships can. Please write “extra-pair” consistently throughout the text.

Response: Thanks, we have written extra-pair throughout the text. 

22. l. 63. Remove “in”.

Response: Thanks.

23. l. 70. characteristics signaling male genetic heterozygosity.

Response: Thanks. 

24. l. 80. Write MHC-I (or MHC class I) and MHC-II consistently throughout the text.

Response: Thanks, we have written MHC-I throughout the text.

25. l. 90. Well-described.

Response: Thanks.

26. l. 93. Parenthesis opens but doesn’t close.

Response: Thanks, we have corrected this.

27. l. 99. During the breeding season.

Response: We have written during the mating period, as we concretely referred to this period because the incubation period is also included in the breeding season. 

28. l. 100. Females and males.

Response: Thanks.

29. l. 133. 100% ethanol? Add the percentage.

Response: We have added 94% ethanol.

30. l. 151. Remove space before %.

Response: Thanks. 

31. l. 199. Low noise-controlled.

Response: Thanks.

32. l. 220. Differently from Mathematics, in Biology it’s close to impossible to “demonstrate” something as new evidence can change our theories and hypotheses. It’s better to prefer the wording “shown” or synonyms.

Response: Thanks, we have changed it.

33. l. 237. Which kind of alcohol? Ethanol? Pure or diluted? Specify the percentage.

Response: We have added 94% ethanol.

34. l. 252. donors were used.

Response: We have maintained “could be used more than once” as some pairs were used only once.

35. l. 273. Missing space between “X” and “2.5”. Use the “X” symbol consistently (see also l. 278).

Response: Thanks.

36. l. 277. birds used in the trials.

Response: Thanks.

37. l. 290. Any bird preference for.

Response: Thanks.

38. l. 299. Remove “as fixed factor” as you already mentioned it in the methods”.

Response: Thanks, we have removed it. 

39. l. 313. See comment 32.

Response: Thanks.

40. l. 326. See comment 32.

Response: Thanks. 

41. l. 328. Remove the second “used”.

Response: Thanks, we have corrected it.

42. l. 348. [53], it is still.

Response: Thanks

43. l. 375. Investigations.

Response: Thanks.

44. l. 403. Remove space between “extra-“ and “pair”.

Response: Thanks.

 

Response to comment of Reviewer 2

Reviewer #2: This paper tests for evidence of odour-based discrimination in birds based on MHC class I diversity and similarity. As an understudied area of research, this paper makes a useful contribution to the literature, but I have two main concerns I would like to see addressed before recommending for acceptance: presentation of methods and data, and the focus and scope of the discussion. I discuss in more detail below.

Presentation of methods and data:

It is essential to present at least a summary (e.g. mean +/- SE) of the MHC class I diversity and distances of the stimuli presented to the focal birds. Such data are necessary to evaluate the differences in allelic diversity and genetic similarity of the paired stimuli birds were tested with. For example, it is possible that birds did not respond differentially to donor bird odours because differences in MHC diversity +/or similarity were very small – if that may be the case, it is important to discuss whether a detection capability threshold based on MHC diversity/similarity was likely to have been met or not. Relatedly, if the distances and diversity differences were very small, I recommend the authors analyze the subset of maximally different stimuli pairings (using some reasonable threshold determined a priori) to see if the focal birds presented with maximally different pairings responded differently.

Response: We have provided a table in the supplementary material to show the mean +/- SE and range of the MHC class I diversity and distances of the stimuli presented to the focal birds. In the manuscript, we show the results of a binomial test aimed to analyse whether birds preferred or not the focal bird with more diversity or more dissimilar in MHC-I. However, in a preliminary analysis we performed another binomial test to analyse whether birds could detect amino acid variant numbers of potential mates by using a generalized linear mixed model with binomial errors and a logit link function (GLMM). We modelled the probability that birds chose a particular side of the chamber (as a dichotomous variable: left (yes) vs. right (not)) in relation to the sex and the differences in the number of MHC-I amino acid variants, and in the body condition, between the two scent-donor birds. We included the interaction between the sex and the difference in amino acid variant numbers in the model to test whether males and females differed in their preferences for partners with different numbers of variants of MHC-I amino acids. We also included the pair of donor birds in the model as a random factor to control for the fact that pairs of donors could be used more than once. We included in the initial model a variable reflecting whether the experimental bird left the chamber when we opened the doors or after 1 min as fixed factor. As this variable was not significant (P > 0.49 in all cases), it was removed from the final models. We performed similar analysis to assess whether the difference in functional numbers, or the difference in amino acid distance or in functional distance influenced the choice of birds. Results of this analysis provided the same result, i.e. birds were not exhibiting any preference, so we decided (as previous referees suggested) to use a simpler analysis. We have also provided these analyses in the supplementary material. We have explained in the discussion: “Our results are robust because we used a large sample size and because we used scent-donor birds of similar body condition in each trial to exclude any potential effect of body condition of scent-donor birds in the choice of focal birds. Also, the large sample size used may guarantee that both the MHC-I diversity and similarity distances were likely appropriate for olfactory detection by house sparrows. Furthermore, we performed preliminary statistical analyses using the difference in MHC-I diversity between scent donour partners or the difference in distance between the scent donour partners the focal birds and we observed that the difference in diversity or distance did not explain the choice of birds (see supplementary material). Therefore, we can exclude that differences in MHC-I diversity or similarity were very small and therefore, house sparrows were not able to detect them.”

Given that these were post hoc tests, it seems that these behavioural trials were not originally designed to test MHC-based odour preferences. The authors should be clear about this and state how and when these data were originally used. I don’t necessarily think it’s inappropriate to evaluate the data post hoc, but I think it is important to be clear and up front about this, and to clearly reference any other papers in which these data have been previously used.

Response: In fact, the experiment was designed to test whether house sparrows can use olfaction to assess the number of MHC-I alleles of potential partners. Griggio and collaborators (2011) published the results of a study that showed that females with low allele numbers in MHC-I preferred males with greater allele numbers. So, we aimed to test whether olfaction may be involved in the detection of allele numbers. We captured the birds and determined the allele numbers, using molecular weight, as described in Griggio et al. (2011). Then, we offered the birds the scent of two potential partners differing in the allele numbers that we obtained. Unfortunately, there was a lab mistake in the assessment of allele numbers, that we discovered after performing the behavioural tests, when the breeding season was finished, and we already released the birds. So, we correctly analysed the allele numbers using the molecular weight and tried to publish the results, analysing a posteriori the number of alleles, because the previous analysis was wrong and therefore, the numbers did not match. When we submitted to ms to try to publish it, referees said that the number of alleles calculated using the molecular weight, following Griggio´s paper, was not longer good, so we performed the metabarcoding analysis and we recalculated the number of alleles using the metabarcoding data. In that way, we obtained the allele numbers as well as the functional distances of the conspecifics offered to the focal birds. As we could not use these data when we performed the behavioural experiment, we had to do the analysis a posteriori. We did not explain this in the text, as we think it does not provide relevant information to the readers to understand the ms, but we have now included in the methods that: “For methodological problems, we could not perform the MHC-I characterization before the behavioural tests. The MHC-I characterization was done a posteriori and therefore, we could not pair scent donour birds in relation to their MHC-I diversity or dissimilarity to focal birds.” 

Focus and scope of discussion:

Related to the methods and study design, the authors should discuss limitations of the study. For example, with respect to whether the MHC diversity and similarity distances were likely appropriate/sufficient for detection by house sparrows.

Response: Please, see previous response related to this issue. We have added this explanation to the discussion: “The absence of odour-based preference for more diverse or more dissimilar potential partners could hardly be explained by a lack of olfactory abilities in house sparrows because house sparrows are known to use olfaction in social contexts [Fracasso et al. 2019], so a low olfactory capability is also not likely to explain this lack of preference. Our results are robust because we used a large sample size and because we used scent-donor birds of similar body condition in each trial to exclude any potential effect of body condition of scent-donor birds in the choice of focal birds. Also, the large sample size used may guarantee that both the MHC diversity and similarity distances were likely appropriate for olfactory detection by house sparrows. We performed preliminary statistical analysis using the difference in MHC-I diversity between scent donour partners or the difference in distance between the scent donour partners the focal birds and we observed that the difference in diversity or distance did not explain the choice of birds (see supplementary material). Therefore, we can exclude that differences in MHC-I diversity or similarity were very small and that, house sparrows were not able to detect them.”

Currently, the authors spend a lot of time discussing the validity of using first choice, and of tapping on the maze after 1 min to force a choice, but they do not discuss the possibility that this method may not be as appropriate for passerines or sparrows in particular, compared to other species in which first choice is more commonly used, such as seabirds, which may reasonably be expected to behave differently in the test chamber than sparrows do. The authors argue that many studies use this method, but they do not point out that at least as many studies (if not more) use cumulative time spent with a sample type, also with good success (especially for passerines). 

Response: We have tried to explain in detail the validity of using first choice, also for Passeriformes, and the limitations of this measure. However, we used live birds as scent donour sources, and therefore, it was not possible for us to assess the time they spent close to the stimuli. We have explained this possibility in the discussion: “Another possibility to explain the lack of preference observed in our study is that a methodological artifact may have masked a preference. In the study with song sparrows, Grieves and collaborators analysed the percentage of time spent close to the two stimuli after 5 minutes from the beginning of the trial. Neither the first choice nor the preference of birds during the first 5 minutes of the trial was reported. We in contrast, analysed the first choice of birds after 5 minutes of exposure. The validity of first choice as a measure of the interest of birds, in particular chemical stimuli, has been previously shown [50, 51, 54, 55, 56], including MHC related scents [15]. However, first choice is a good measure of the spontaneous interest of an animal to a particular cue, whereas time spent close to the stimulus [57] may rather be related to the behaviour shown later on in the series of events to a certain cue. Using life birds as scent donors, the first choice was a valid measure of the response of birds to scent in our study. However, to analyse whether the lack of preferences for the scent of most MHC-I diverse or dissimilar conspecifics is maintained over time, more studies are needed to assess the subsequent response of birds to the scent of potential partners, for instance using uropygial gland secretions as scent sources.”

Finally, there is no discussion of how the timing of the experiment and the physiological condition of the birds could have impacted results or predictions of focal bird preference. In the Leclaire study referenced, birds were incubating and may have preferences related to this, whereas in the Grieves study birds were in breeding condition and may have been motivated differently than in the Leclaire study. This should be included in the discussion alongside consideration of the breeding condition of the birds used in the author’s present study and how this may have affected results and/or predictions/expected outcomes.

Response: We explained in the methods section that we performed the experimental study during the mating period, i.e., during the breeding period with birds that were not paired with birds from the opposite sex and that were not laying or incubating, so they may be in breeding conditions. We now have remarked this point the discussion: “Differences between our results with house sparrows and those with blue petrels can be due to differences in the breeding conditions of birds between those studies, because Leclaire and collaborators tested blue petrels that were incubating [15] and therefore, not searching for a partner. In contrast, we performed the experiment during the breeding period with birds that were separated by sex, and therefore, they were not paired or incubating, so, they may be interested in searching for a partner. Differences between our results and those obtained with song sparrows cannot be attributed to differences in the breeding condition of birds, as in both studies birds were tested during the breeding period and were not incubating.”

Detailed comments:

Line 86: why would MHC I be related to bacterial loads?

Response: We do not know why the authors obtained that result, it may have been an unexpected finding. We have removed the reference. 

Lines 93-101: odd to name authors of one study but not the other, and name the study species in one study but not the other. Suggest rephrasing both sentences for consistency of results presentation. It is also worth explaining here or in the discussion the different predictions that can reasonably be made for each sex at different breeding stages (such as during mate choice and pairing vs during incubation) or different predictions that might be made for different taxonomic groups (seabirds vs songbirds) that may differ in level of parental care and investment between the sexes (sex roles, timing of care/investment for each sex, etc).

Response: We have modified the paragraph to be consistent in the use of author names and species names. We have also specified that: “In this study we have experimentally explored whether olfactory signals play a role in the preference for potential partners with greater MHC-I dissimilarity and/or diversity in house sparrows, Passer domesticus, during the mating period.” 

Line 101: results of Leclaire and Grieves are conflicting but are they controversial? Perhaps not, especially when considering time in breeding cycle as described above.

Response: “Thanks, we meant conflicting, we have rephrased”

Line 115: MHC I, II, or both?

Response: We have now added MHC-I. 

Line 120: high or intermediate MHC diversity

Response: Thanks, we have added intermediate.

Line 133: when was the genotyping performed? Birds were captured Feb-Mar 2015 but it’s not clear when genotyping was done in relation to when the behavioural trials were conducted.

Response: We have explained that: “For methodological problems, we could not perform the MHC characterization before the behavioural tests. The MHC characterization was done a posteriori and therefore, we could not pair scent donour birds in relation to their MHC-I diversity or dissimilarity to focal birds.”

Line 136: given that all birds were initially allowed to interact in a group aviary, could it be that mate choice had already taken place, or preferences have otherwise been established? Should discuss whether/how this may have impacted results. Was any evidence of courtship or breeding activity observed during the time birds were group housed?

Response: We have now tried to explain better that birds were located in aviaries and separated by sex, i.e. there were aviaries containing only males and aviaries containing only females, therefore, no reproductive activity could be performed by birds. We have commented on this in the discussion: “Previous experience with scent donour birds may have masked a preference for MHC-I diverse or dissimilar partners. However, we captured birds during February and March and kept females separated from males. Birds were separated by sex in different aviaries, to avoid establishing any interaction with potential partners that may have affected olfactory preferences. Therefore, neither the timing of the experiment, previous interactions during the breeding period or the physiological condition of the birds may explain the lack of preference for MHC-I dissimilar potential partners.”. 

Line 143: include primer references/original source for primer development

Response: We have added references, where the primers were designed and modified. 

Line 144: please describe what is meant by 'illumina sequencing primer sequences' in more detail so methods can be evaluated.

Response: These Illumina primers (adaptors) are used for allowing the fragments to attach to the cell flow. Now it is explained in the text.

Line 148: please explain how index sequences differ from illumina sequencing primer sequences and why these were attached in a separate round of PCR

Response: These indexes are short sequences (usually around 6 base pairs combinations), unique for each specimen (now indicated in the main text). Two-step PCR is a convenient method to generate amplicon libraries for Illumina sequencing, as it was shown in multiple studies (see for instance, Cruaud et al. 2017. Scientific Reports 7: 41948).

Line 149: demultiplexing?

Response: We are referring here to the possibility of analysing multiple specimens together. The demultiplexing step is posterior.

Line 158: what criteria were used to set this 1% threshold? Overall, more information on sequence data processing methods is needed to properly evaluate these methods. The only reference given is 10 years old and, given the rapidly changing nature of data handling methods in this field, it is essential to ensure the most up to date and appropriate methods are being used for this type of data.

Response: The 1% threshold is still adequate as a first filter to eliminate artefactual alleles. Studies such as Drake et al. (2022) (now cited) support this threshold. We have also added the reference of Razali et al. (2017), which explains the cleaning of the raw data in detail.

Line 159-160: rarefaction is not appropriate for these data (see McMurdie, P.J. and Holmes, S., 2014. Waste not, want not: why rarefying microbiome data is inadmissible. PLoS computational biology, 10(4), p.e1003531.). In Strandh et al. 2012, this 200 cutoff was based on inclusion of technical duplicates, which his reliable only within the sequencing run being evaluated. Were technical duplicates included in this study? If so, this procedure must be described and explained. If not, this method is inappropriate and should not be used. Instead, please update with more recent/current methods.

Response: We did not use duplicates in our data, but we counted on blanks. The number of reads per sample obtained indicated that 200 reads could represent spurious results. For that reason, we have considered this figure as adequate.

Line 169-170: JTT SMS explain/write out in full on first mention

Response: JTT stands for the initials of the authors' names who described this model. We have added the full name and the reference. We have also indicated the meaning of SMS.

Line 201: was the light turned on or was it kept dark throughout the trial?

Response: We have explained that: “Donor birds were kept in darkness during the whole trial duration and reduced space, preventing them from moving or calling.” 

Line 217: was experimenter present the entire time? Was experimenter visible to the focal / test birds? Was the test chamber opaque or transparent?

Response: The experimenter was present during the entire trial, but not visible/audible to the focal bird, as the chamber was opaque to maintain birds in darkness.

Line 224: admittedly, I struggle to accept this method as being appropriate... Regardless, it is important to present a summary of how many birds was this done for. A 1 min trial duration seems very short. Many similar studies allow 15 min for birds to choose, even those scoring first choice. I recognize you cannot change this now, but in future, why not use a longer trial duration and evaluate both first choice and choice / time?

Response: We have now explained in the methods that: “68 birds made their choice within less than 1 minute after opening the doors and 33 birds afterwards.” Only one minute may seem to be a short time but, in preliminary trials, we left birds for longer during trial duration and in some cases birds would not move for more than 30 minutes. So, it seems that most birds chose as soon as doors were opened but the ones that did not choose at that moment, spent a long time before moving. Previous results showed that knocking at the door did not influence bird choice, likely because the initial choice of birds was merely reenforced but nor influenced (Amo et al. 2012a,b, 2015). Results of this study showed that knocking at the door did not affect bird choice. Therefore, for ethical reasons, we decided not to unnecessarily increase the time of birds inside the chamber, because not only focal birds were in the chamber, but also scent donour birds. 

Line 232: why were only 120 of the original 151 birds tested?

Response: The rest of birds were used in a different behavioural experiment. 

Line 235: please include discussion of the side bias exhibited by test birds. How often was the more vs less diverse and more vs less similar individual placed in the left or right side of the olfactometry chamber?

Response: We have explained in more detail that: “Focal birds made 61 % of their first choices to the left and 39 % of the choices to the right side. There were not significant differences in the number of scent donour birds more diverse or more dissimilar between those located in the right side or in the left side of the aviary (GLZM, P > 0.30 in all cases). In relation to the amino acid diversity, in 47 trials the most diverse bird was located in the right side of the olfactometry chamber and in 50 trials in the left. In relation to the functional diversity, in 49 trials the most diverse bird was in the right side and in 47 trials in the left. In relation to amino acid distance, the most dissimilar bird was in the right side in 56 trials and in the left in 63 trials. In relation to the functional distance, in 54 trials the most dissimilar bird was in the right side and in 65 trials in the left.”

Line 237: what type and % of alcohol was used?

Response: We have explained that we used “94% ethanol”

Line 266: what were the distances? How was it determined that this was a detectable threshold or that differences in allelic diversity and genetic similarity were otherwise reasonable? Must present data on what MHC distance and allelic diversity pairings were used.

Response: We have provided this data in supplementary material. Please, see the explanation below about the statistical analysis using differences in diversity and distance to explain the bird choice.

Line 286: these trials appear to have been used for multiple experiments/to test multiple questions post hoc. If so, this should be clearly stated somewhere in the methods.

Response: As we explained before, we did not use these trials in other experiment nor to test multiple questions post hoc. We have now explained that: “Due to methodological problems, we could not perform the MHC characterization before the behavioural tests. The MHC characterization was done a posteriori and therefore, we could not pair scent donour birds in relation to their MHC-I diversity or dissimilarity to focal birds.” 

Line 316-321: again it's odd presentation to name author in one study and not the other, suggest rephrase so both are consistent.

Response: Thanks, we have corrected it.

Line 321: song sparrows are relatively asocial, blue petrels relatively social, and house sparrows relatively social, but all species may differ in level of sociality during breeding – please discuss how you think sociality may be involved.

Response: Despite that the three species may differ in sociality during the breeding period, we hardly believe that differences in sociality may explain the lack of preference for MHC-I dissimilar partners in our study. We have tried to improve the discussion to implement this potential explanation in our results.

Line 326: many of these studies set a maximum trial duration of 15 min, presumably to allow time for the birds to choose on their own. The present study seems to allow the birds only 1 min, followed by tapping on the test chamber to force the birds to move, which while use in prior studies of this group. The potential impacts of this should be discussed.

Response: Birds were exposed for 5 minutes to the scents, not only one minute. Therefore, if they did not choose during the first minute, the trial duration was extended to 6 minutes in total. Despite that most birds chose within the first minute, we included a variable reflecting whether birds chose within the first minute or after and it did not influence the results.

Line 327: but what MHC differences were used in terms of both diversity and dissimilarity? Again, this information needs to be provided.

Response: We have provided in the supplementary material the means, the data, as well as the analysis of the choice of birds in relation to both the differences in diversity and dissimilarity.

Line 331-341: this would be a good place to discuss the results in terms of the relative differences in MHC I diversity and similarity of the stimuli, and how this may have affected the results. 

Response: We have added a paragraph in the discussion. Please, see lines 272-282.

Line 342: returning back to a justification of the first choice method is jarring here; what does this have to do with future work or using preen oil instead of whole body odour? Why would preen oil be superior to use of whole body odour? Suggest restructuring the relevant paragraphs for clarity/flow.

Response: We have tried to clarify it: “Using life birds as scent donors, assessing first choice is a valid measure of the response of birds to scent in our study, because we had to ensure scent-donor birds remain in silent during the trial to avoid acoustic cues that may have affected focal bird choice. However, further studies are needed to assess whether the lack of preference for the scent of most MHC-I diverse or dissimilar conspecifics is maintained over time, for instance using uropygial gland secretions as scent sources.”

Line 347: potential mechanisms for scent-cues of MHC genotype would be better placed in the introduction to set up the study rationale. Since the study did not find evidence for odour choice, I don’t think this discussion of potential mechanisms needs to be here.

Response: Thanks, we have tried to improve this paragraph. We have maintained it in the discussion as we argue that it is needed to perform additional research to examine the mechanisms that may allow MHC-I discrimination in birds.

Line 353: birds lack a VNO – what is the relevance of including this information here?

Response: We have now tried to better explained this.

Line 354: would MHC I be expected to be involved in influencing bacterial profiles?

Response: Thanks, we have tried to clarify.

Lines 357-360: please put this information in the context of the present study.

Response: We have tried to improve the whole paragraph to put the information in the context of our study.

Line 374: a lot of time is spent early in the discussion justifying the use of first choice, but now seems to conclude by saying that different methods should be used - what methods should be used/are recommended? Isn't it worth suggesting that choice over time (cumulative time spent with stimulus) might be a good approach to consider for this type of research in songbirds? First choice has been used successfully in several studies, but so has cumulative time spent, and this should be acknowledged and evaluated. Overall, I think the discussion should be reorganized and more tightly focused around the research question and study results.

Response: Thanks, we have tried to improve the discussion of our results.

---

## [Decision Letter · Decision Letter 1]

12 Aug 2022

PONE-D-22-00952R1House sparrows do not exhibit a preference for the scent of potential partners with different MHC-I diversity and genetic distancesPLOS ONE

Dear Dr. Amo,

Thank you for submitting your manuscript to PLOS ONE. After careful consideration, we feel that it has merit but does not fully meet PLOS ONE’s publication criteria as it currently stands. Therefore, we invite you to submit a revised version of the manuscript that addresses the points raised during the review process.

We look forward to receiving your revised manuscript.

Kind regards,

Magdalena Ruiz-Rodriguez

Academic Editor

PLOS ONE

Reviewers' comments:

Reviewer's Responses to Questions

**Comments to the Author**

1. If the authors have adequately addressed your comments raised in a previous round of review and you feel that this manuscript is now acceptable for publication, you may indicate that here to bypass the “Comments to the Author” section, enter your conflict of interest statement in the “Confidential to Editor” section, and submit your "Accept" recommendation.

Reviewer #1: (No Response)

Reviewer #3: (No Response)

2. Is the manuscript technically sound, and do the data support the conclusions?

Reviewer #1: Yes

Reviewer #3: Partly

3. Has the statistical analysis been performed appropriately and rigorously? 

Reviewer #1: Yes

Reviewer #3: Yes

4. Have the authors made all data underlying the findings in their manuscript fully available?

Reviewer #1: Yes

Reviewer #3: Yes

5. Is the manuscript presented in an intelligible fashion and written in standard English?

Reviewer #1: Yes

Reviewer #3: Yes

6. Review Comments to the Author

Reviewer #1: The authors exhaustively addressed my concerns and I’m quite satisfied of the work done. However, they forgot to replace body condition with body weight throughout the text (see my only major comment). I added some minor comments to this revised version. Line numbers refer to the manuscript with tracked changes. Finally, I suggest to include the line numbers referring to the changes that were made to the text next time, thus facilitating their revision from the reviewers.

Major comments

1. The authors state (authors’ response 7) that they have replaced body condition with body weight in the text after considering my comment. However, I still see several mentions to body condition throughout the manuscript (ll. 248-249, 250, twice in l. 397). Please change them as well.

Minor comments

2. ll. 59-60. The paragraph has greatly improved. However, I would remove the last part of the sentence (…but indeed suggest additional…) as it is repeated (in a better way) in the following sentence.

3. l. 68. The parenthesis closes but it is never opened.

4. l. 77. Remove comma after “Although”.

5. l. 245. …lasted up to…

6. l. 360. So, we expected birds would have been…

7. Table 1 and 2. The tables are now easier to read but I think the authors can go even further. They can physically link table 1b and 2b below table 1a and 2b respectively if in the top left cell of table 1a, 1b, 2a, 2b (now empty) they add the letter and a short title, e.g. for table 1a: a) amino acid diversity.

8. l. 262. What does GLZM stand for? Do you mean GZLM? Even so, I very rarely came across this abbreviation before.

9. l. 385. …live/living birds…

10. l. 387. …they remained silent during…

11. l. 378. …after 5 minutes of exposure to the birds’ scent during which no choice could be made.

12. ll. 220, 251, 254, 260, 366, 401, 402, 781, 794. scent-donor …

13. l. 384. …between the scent-donor partners and the focal birds, and…

14. Some sentences in the discussion are a bit convoluted. I believe a text revision from an external (mother tongue) English speaker would help to make the text more fluent.

Reviewer #3: (No Response)

7. PLOS authors have the option to publish the peer review history of their article (what does this mean?). If published, this will include your full peer review and any attached files.

Reviewer #1: No

Reviewer #3: No

---

## [Author Response · Author response to Decision Letter 1]

21 Oct 2022

This study adds to the expanding field of avian chemical communication by studying the olfactory preferences of house sparrows as they relate to MHC-I genes. The authors find no preferences in either males or females. I think this is first study to look at MHC-I genes and olfactory preferences in birds, so this manuscript represents an important contribution to this growing field. I thought the writing could be benefit from some re-organization and editing to improve clarity and increase readability as I found it difficult to follow in certain places. I have tried to provide extensive comments to help identify the places where I think some changes would be beneficial. I also have several concerns relating to the decision to analyze continuous genetic distance/dissimilarity variables as a dichotomous categorical variable, and some questions about interpretation of the results, particularly the new supplementary results that were added in the prevision round of revisions. I have divided by review up with comments or questions for the various sections of the manuscript.

Response: Thank you very much for the detailed and extensive comments. They were very helpful, and we have tried to implement them to improve the manuscript. Please find an explanation/response to each of your comments. 

Introduction

I think the introduction could be more streamlined and the organization could be improved to better convey the important points. I have provided multiple suggestions below.

Paragraph 1:

This paragraph starts out discussing female choice and then makes a diversion to male choice in red jungle fowl in the last sentence. This currently reads awkwardly. Since the experiment described in this manuscript tested the ability of both males and females to discriminate MHC dissimilarity/diversity in opposite sex individuals, I think this entire paragraph would be stronger if it discussed mate choice as a process where both males and females may participate. This is a reasonable assumption for house sparrows (and many bird species) where there is some degree of biparental care. While females might be more motivated to be choosey because they invest more in reproduction, males are also contributing to parental care and therefore may also participate in mate choice. I assume the authors agree otherwise they would not have bothered to test the olfactory preferences of males. Lastly, a study in birds that documented MHC-based mate choice by males (for MHC Class II) is Hoover et al. 2018 Molecular Ecology (https://onlinelibrary.wiley.com/doi/abs/10.1111/mec.14801). This could accompany the red jungle fowl citation to help strengthen the discussion of MHC-based mate choice by males.

Response: Thank you very much for your suggestions. We have added the reference. However, we have maintained the order trying to first emphasize the hypothesis of females being the choosy sex in mate choice. Our reasoning here was to first explain the choice-making of females, and later we have explained that “Furthermore, molecular techniques have revealed that males also select mates”, adding the reference you suggested.

Paragraphs 2 and 3:

MHC is not explained until Line 78, halfway through paragraph 3, but there are multiple references to MHC in the second paragraph (and also at the end of the first paragraph). This is not a great approach because it assumes the reader is already familiar with MHC. I recommend that the information from Lines 78 through 84 is presented a lot earlier in the introduction to avoid confusion.

Response: Thank you very much for these comments. We have now explained MHC functions earlier in the introduction. 

The text in these paragraphs repeatedly uses the terms “genetic dissimilarity” and “heterozygosity” but it is unclear if this refers specifically to MHC genes or more widely to genome-wide heterozygosity or genetic dissimilarity at other loci. There are several studies from birds that have documented covariation between non-MHC genetic markers and odor profiles (see Leclaire et al. 2012 Proc B https://royalsocietypublishing.org/doi/10.1098/rspb.2011.1611 and Whittaker et al. 2019 Animal Behaviour https://www.sciencedirect.com/science/article/abs/pii/S0003347219300831). It should be clarified whether the studies included here are specifically relating to MHC genes or not.

Response: We have now added in the respective paragraphs that we refer to MHC-related genetic characteristics.

Paragraph 4:

Line 102. It is unclear to me why the findings of Leclaire et al. and Grieves at al. are being described as “controversial”. Is this word used because many people assume birds are unable to use odors in social contexts? I think this choice of word could come across as negative towards these past studies and I do not believe that is not the intention of the authors. I recommend the wording is changed here.

Response: Thank you for your comment and pointing out that this could be misread. We have now changed the wording as it was not our intention to imply any criticism of prior studies but to merely point out the opposite findings between prior studies and our findings. We have changed “controversial” to “opposite” results in the case of female preferences.

In my opinion, there are three pieces of evidence that work together to support olfactory mediated MHC-based mate choice in birds (or any vertebrate species):

1. Observations of non-random mating associated with MHC genes. There are a decent number of published studies in birds.

2. Studies showing that chemical profiles that contain information about MHC genes. In birds, this has been shown for preen oil (Leclaire et al. 2104 Scientific Reports https://www.nature.com/articles/srep06920, Slade et al. 2016 Proc B https://royalsocietypublishing.org/doi/10.1098/rspb.2016.1966#sec-7) and plumage (Jennings et al. 2022 Proc B

https://royalsocietypublishing.org/doi/full/10.1098/rspb.2022.0567).

3. Behavioral experiments where individuals exhibit olfactory preferences for individuals based on their MHC genes.

This study clearly falls into the third category, but all three pieces are needed to make a strong case for MHC-based mate choice via olfaction. The introduction of this manuscript has several citations for MHC-based mate choice and there is a discussion of the behavioral evidence in the 4th paragraph. I think it would strengthen the introduction to briefly touch on the evidence from chemical profiles. This may help to convince anyone skeptical about the avian sense of smell that this area of research deserves more attention. I also think this is relevant to the non-significant results obtained in this study. One of the outstanding questions that remains based on the findings of this manuscript is whether or not house sparrow body odor contains information about MHC-I. One possibility is that their body odor does not reflect information about MHC-I.

Alternatively, information about MHC-I may be present, but the birds either do not detect it via olfaction or do not use it in the context of mate choice. These points could be easily added to the discussion if this idea had already been introduced in the introduction.

Response: Thank you for your comment. We do not think that previous studies about observations of non-random mating associated with MHC genes support that avian olfaction is involved in olfactory recognition of MHC characteristics of mates, as other cues may be used and are not excluded in those studies. Indeed, we agree that previous studies showing that chemical profiles vary in relation to MHC genes support the idea that olfaction can be a mechanism used by birds as well as other animals. We have included this idea in the introduction, as well as in the discussion. Thank you very much. 

Methods

I had a number of questions about the details of the preference tests. I have grouped all of my comments together.

Timing of each trial:

I found the protocol for the behavioral trials difficult to follow. I believe there was a 5-minute acclimation period during which the focal bird was exposed to the odors from the two donor birds. After 5 minutes, the two trap doors were opened by the human observer and the focal bird was given one minute to make a choice. If they did not make a choice after 1 minute, the human observer tapped on the arena and the side the bird moved to following the tap was noted as their choice. The total duration of the experiment was 6 minutes per trial. I think this is correct, but it took me reading the methods section several times to get all this information. I struggled to understand the protocol because the section about the “Behavioural Experiment” jumps around a lot. The reader learns about the 5 min acclimation period in Line 205, but the 1 min period during which the bird is allowed to make a choice is not discussed until Line 228, almost an entire paragraph later. It is easy for the reader to lose track of what is going on. Interspersed between these two protocol details is a lot of information about the design of the experimental arena. I recommend that this section is restructured so that the design of the arena with its various compartments, fans, how the focal bird received the odors etc is provided first. Then, move into the specifics of the protocol for the experiments with the different timings (5 mins, 1 min, tap on arena etc).

Secondly, it reads to me like all individuals in the study made a choice. That is often not the case for olfactory preference tests where many protocols remove the individual after a certain amount of time and mark the trial as a “no choice” result. If would be helpful if the methods stated that the tapping always resulted in the bird selecting a donor bird. Somewhere around Line 231 seems like a logical place for this information.

Response: Thank you for your comments. We have modified this section.

Size of cages:

Please add the dimensions of the two little cages discussed on Line 207 where the donor birds were housed.

Response: We have provided the dimensions of the cages.

Visual cues in the experiment:

I believe all the birds (donor and focal) were in the dark to eliminate visual information. Line 205 says “the bird was maintained in the dark for 5 min” and Line 216 says “donor birds were kept in darkness for the entire trial duration. Lastly, on Line 222 it says, “the chamber was opaque to maintain birds in darkness”. This all seems fine except I am unsure how the human observer who scored each trial was able to see which donor bird the focal bird selected under these conditions. Was the light in the center chamber turned on after the 5 min acclimation period? How exactly did this work? Was the observer able to hear where the bird moved and did not need to make a visual observation? Please add details to the methods to clarify.

Response: Thank you for your comments. We have now explained in the methods that the experimenter could hear the choice of the bird and confirmed it by introducing a hand though the central chamber.

Pairing of donor birds based on morphometrics:

The two focal birds used in the trials were matched for a number of visual characteristics including body size, weight, and size of badge. As the focal bird was unable to see the donor birds, I assume this was done to remove any potential correlation between body condition/dominance status and avian body odor? I believe the intention was to have MHC be the primary factor that differed between the two donor birds so that any differences in odor profiles could most likely be attributed to MHC and not another confounding factor. If this is correct, I think it would be beneficial to clarify this in the paragraph that begins on Line 233.

Otherwise, it is not obvious why so much care was taken to match visual traits for the donor birds in an experiment that only examined olfactory preferences.

Response: Thank you for your comment. We have added an explanation in lines 258-264.

Data Analysis

I have a number of questions relating to the data analysis. If I had to replicate this analysis, I would have a hard time doing so because I am not entirely sure how each of the models are actually different. In certain places I have specific questions or comments. Throughout my comments, I have done by best to explain what I think was done. I hope this will help the authors make edits to clarify their methods, particularly in places where I have misunderstood the analysis.

I think there are two ways to examine the behavioral data. Each trial has a triad of individuals; one focal bird who performed the experiment and two donor birds that were used as scent sources. One set of analyses could compare the MHC genotypes of two donor birds to each other (ignoring the genotype of the focal bird). This could be done by measuring the number of amino acids encoded by each donor genotype or by obtaining a measure of functional diversity for each donor genotype. This type of analysis would determine if there was a preference by the focal bird for individuals with higher MHC diversity. A second set of analyses could examine the MHC genotype of the donors relative to the MHC of the focal bird. This approach differs from the first because it considers the genotype of the focal bird, which is important as individuals may be looking for a mate with a genotype that complements their own rather than simply preferring a mate who has maximum genetic diversity (a complementary mate vs a genetically diverse mate).

This analysis would require measuring the genetic distance between focal and donor birds, which could be based on amino acid differences or functional differences. This analysis would test whether the focal bird more often selects an individual who has more dissimilar MHC relative to their own genotype. I believe both these approaches were implemented to some degree in this study, although I am not entirely unsure.

Response: We always considered the MHC characteristics of the focal birds in the analysis. In a first set of analysis, we analysed whether the choice of focal birds was influenced by the diversity, both in the amino acid alleles as well as in the functional alleles of scent-donor birds, considering the diversity of MHC alleles of focal birds. MHC diversity of focal birds was included as an independent variable in the analysis. In a second set of analysis, we analysed whether the choice of focal birds was influenced by the dissimilarity in MHC alleles of scent donor birds. To estimate similarity (or dissimilarity) we calculate the distance between each scent donour bird and the focal bird. We have revised this section in the methods part trying to clarify the analysis: “We performed two sets of analyses to examine scent preferences of focal birds for diversity or similarity of potential partners, always considering the MHC of focal birds in the analysis. We performed a first set of analyses to examine the preferences of the focal birds for the diversity in amino acid alleles and functional alleles of scent donor birds, considering MHC diversity of the focal bird….. In a second set of analyses, we analysed the preference of the focal bird for the MHC similarity of scent donor birds, by using the genetic and functional distances between the scent donor birds and the focal bird. We analysed the preference of the focal bird for the MHC similarity of scent donor birds, considering the similarity between the focal bird and the scent donor birds.” 

There are four main analyses described. I have done my best to describe what I think is going on in each analysis. 

The first analysis is described in the paragraph starting on Line 261. It used amino acid variant number. Based on the information given in Line 266, it reads as if the number of amino acid variants of each donor bird was compared to the number of amino acid variants in the focal bird, and the two donors were classified as having either higher or lower numbers of amino acids than the focal bird. The model tested the probability that the focal bird preferred the donor with the higher number of amino acid variants than the donor with the lower number of amino acid variants. This type of analysis is similar to the second option I described above because it considers the genotype of the focal bird. I have two comments related to this analysis:

- Please give a definition or description for what “amino acid variant numbers” means. Is this the number of amino acids encoded by their various MHC alleles?

Response: We have revised this section in the methods part: “We calculated MHC amino acid diversity for each individual as the number of unique amino acid sequences. The amino acids in the particularly polymorphic peptide-binding regions of the MHC molecules determine what antigens can be bound and are therefore crucial for the function of each allele. We calculated functional diversity for each individual coding the amino acid sequences according to the chemical binding properties of the amino acids (Strand et al. 2012; Leclaire et al. 2017; Lucask et al. 2017).”

- As the MHC of the focal and donors was unknown during the experiment, how many trials occurred where both of the donor birds had a higher amino acid variant number compared to the focal bird? Similarly, both of the donor birds had a lower amino acid variant number compared to the focal bird? Another way of putting this is, how many trials did you have to exclude from your analyses after you obtained the genotype information?

Response: Thank you for your comments. We have included the missing explanations in the methods sections (lines 276-284): “Since MHC determination was performed a posteriori in the analysis of differences in the number of the difference in amino acid diversity we excluded 22 trials because difference in the amino acid diversity was 0. Therefore, for these trials it was not possible to assess preference based on this analysis. In the analysis of difference in the number of functional diversity, we excluded 23 trials because the difference in the functional diversity was 0.”

A second analysis is described in the paragraph starting on Line 275. It looks at the number of MHC functional alleles. This model includes the number of functional MHC alleles for the two donor birds and the number for the focal bird (I assume as a covariate), but it does not measure the difference in functional alleles between the focal bird and each of the donor birds. I think this is similar to the first type of analysis I described above.

Response: When analysing diversity, we included the MHC-I diversity of the focal bird as a covariable in the analysis, both when we analysed amino acid diversity and functional diversity. We analysed whether the focal bird selected (or not) the scent of a potential partner with a more diverse MHC-I, considering the MHC-I diversity of the focal bird as a covariable.

Lastly, two other analyses are described in the paragraph on Line 282. Both seem to be similar to the second type of analysis I have described above because they consider the difference in MHC between focal and donor birds. The first measures this difference using MHC-I amino acid distance and classifies each donor as having higher or lower. For example, if the amino acid distance between focal and donor 1 was 4 and the distance between focal and donor 2 was 18, the focal-donor 2 pairing would be assigned “higher” and focal-donor 1 pairing would be assigned “lower”, I think? Similarly, a second analysis was done in the same way except the genetic distance was measured as functional distance. If this is not correct, please consider making edits to clarify. 

Response: Yes, as you understood correctly, this was our approach for the analysis. 

Also, I have several specific questions:

- I am unclear how amino acid distance is different from the amino acid variants that was used in the first analysis. Possibly a better description of amino acid variants as I requested above would help clarify this.

Response: We have tried to clarify the term amino acid variants. Amino acid distance is the genetic distance between the scent donor bird and the focal bird considering amino acid alleles.

- Line 287: “in relation to the sex of the focal bird”. Why is the term “sex” used here? 

Response: We have now explained that “We included the sex of birds in the analyses to examine whether the sex of the focal bird influenced the preference of birds.” 

Line 290: please state the version of R that was used (e.g., 4.2.1) and double check the citation information. The current citation says 2012, which seems incorrect as I suspect the authors used a more recent version of the software for their analyses.

Response: Thank you very much for pointing this out. We have now added the correct reference for the R version that was used in our study.

I have a general question that relates to the analyses in the main text. Why were the data turned into a dichotomous variable of lower or higher? It seems that for each of the genetic distance/dissimilarity measures, there should be actual numbers (aka continuous variables) that describe the magnitude of the difference between different individuals. I think the analysis would be far more powerful if the values were kept. By converting to higher/lower you are treating trials as similar that could have been quite different. Here is an example:

Trial 1: distance between focal-donor 1 is 2, distance between focal-donor 2 is 3. In this example, focal-donor 1 would be classified as lower and focal-donor 2 as higher. In reality though, there is very little difference between the pairings, and we may expect that the focal bird would not have very strong preferences in this situation.

Trial 2: distance between focal-donor 1 is 15, distance between focal-donor 2 is 3. In this example focal-donor 1 would be classified as higher and focal-donor 2 as lower. Now there is a very large difference between focal-donor 1 and very little difference between focal-donor 2. If house sparrows are evaluating MHC-I using odors, I expect this scenario could play out differently than above. I think the authors need to better justify their decision to use the dichotomous classification as it is unclear to me why this was done, and it seems to result in a loss of information that make their results less conclusive.

Response: We agree with you that analysing exact differences in diversity and distance may be a more precise analysis. However, in the first version of the manuscript, we reported the analysis using distance measures as continuous variables, but previous referees suggested to use categorical variables to make the analysis easier to interpret. Previously, we used continuous variables, but this complicated the analysis because we were analysing differences between the distances, that is per se not easy to interpret, and results were similar, i.e., birds are not exhibiting a preference for the scent of more MHC diverse or dissimilar potential partners, even when we took the degree of diversity or dissimilarity into account. Therefore, following the advice from the previous reviewers we now show the results of the analysis using categorical variables in the main text. We have now added a supplementary file to show the results of the analysis using continuous variables as well in the current ms. We have revised the Data analysis section accordingly: “In these set of analyses, we considered the choice as a categorical variable, i.e. we analysed whether focal birds chose (or not) the most MHC diverse or dissimilar potential partner, without considering the degree of differences between the diversity or similarity of scent donor birds. The analyses and results considering the differences in diversity or dissimilarity of scent donor birds are provided in the supplementary material.” 

Results

I have two main concerns relating to the results and the current interpretation of these findings.

On Line 245-248 the manuscript says “Focal birds made 61% of their first choices to the left and 39% of the choices to the right side. There were no significant difference in the number of scent donor birds more diverse or more dissimilar between those located in the right side or left side of the aviary”. This seems somewhat concerning. If there was no trend in the placement of more diverse or more dissimilar donors, then a strong preference for the left side of the aviary suggests that the choice made by the focal bird was less about the odor presented and possibly associated with a spatial preference for the left side that was unrelated to the presented odors. The models described in the data analysis do not appear to account for whether the choice was on the left or right side. I think this result needs to be carefully considered.

Response: We have now included the side in the analysis of the main text as well as in the supplementary material and mentioned it in the discussion, although, results of preferences do not change when including the side in the statistical analysis. 

On Line 382 some preliminary analyses are mentioned. The findings from these analyses are given in supplementary Table 1. I think these analyses are similar to the analyses in the main text, but they used the genetic distance measures as continuous variables, which I think is actually a more powerful approach (as I have discussed above). 

Response: In the first version of the manuscript, we reported the analysis using distance measures as continuous variables, but previous reviewers suggested to use categorical variables to make the analysis easier to interpret (see above). We agreed that using continuous variables makes the analysis more complicated to interpret because we analysed differences between the distances. Therefore, following their advice we now show the results of the analysis using categorical variables but adding a supplementary file to show the results of the analysis using continuous variables, providing both analyses to the reader.

This section goes on to conclude “therefore, we can exclude that the differences in MHC-I diversity were very small and house sparrows were not able to detect them” (Line 387). I do not agree with this statement based on the findings presented in supplementary Table 1. In the lower half of this table, the differences between the two focal birds, or the differences between each focal-donor pairing are given. The mean distance for each of these is very low and often very close to zero. My interpretation of these values is that for most of the trials there was very little difference in MHC-1 between all 3 individuals used. 

Response: We do not think difference in the distances are low. As you can see in the table the mean difference between the focal bird and the donor birds was 0.70 + 0.01 from a maximum of 1. So, there were differences between the distances among birds. However, when we subtract the distances between both pair of birds (distance between scent donour left and focal – distance between the scent donour right and focal), the difference of both distances is a small value because some differences are positive and other differences are negative, depending on whether the scent donour bird with more distance is in the right or in the left. If you see the mean and maximum values of differences, minimum values are -0.7 or -0.4 and maximum values are 0.5. Please, take into account that distance between individuals range from 0 to 1.

If this was the case, this could be a very important reason why there were no significant findings in this study, which is the opposite of what is stated in Line 387. We would not expect the birds to exhibit strong olfactory preferences if the two donor birds had very similar MHC and/or the two donors had similar MHC relative to the focal bird. These values either need to be explained more clearly or the authors need to reconsider their interpretation of these values. 

Response: We understand your concern, but the small value is not because distances between the focal bird and the scent donor birds are small, it is because of the difference between both distances. Therefore, we are quite confident that the lack of preferences of focal birds can not be attributed to a lack of differences in the genetic distances between the birds used. Furthermore, due to the large size and that birds come from the same population, differences in genetic distances may be quite realistic, and therefore, we consider that our results are not impacted by low genetic distances between the experimental birds. 

I also have two minor comments relating to Table 1:

- Use decimal places rather than commas for the values in the table.

Response: Thank you for the remark. We have changed this.

- In the middle of the table, two rows have minimum values of -9 and positive values of +9. Is this a type or is this correct? It seems odd that these values would be the same for both amino acid diversity and functional diversity.

Response: The values are ok, these values are maximum and minimum values and it is because there are 3 birds that have 10 amino acid alleles that correspond to 10 functional alleles. 

Discussion

The discussion focuses extensively on the methods of the experiment and how the methods may explain the results. This is fine but I think the paper would be stronger if it devoted a little more space to biological explanations and interpretations of the findings. The paragraph starting on Line 400 is the most interesting and I have provided some additional ideas to potentially expand the discussion.

Line 374: this is a pretty extreme statement. I recommend toning down the language a little.

Notably, the wording “could hardly be explained” is rather strong. I think it is fair to say that house sparrows use olfaction in other social contexts, so it seems probable that odors also play a role in mate choice. However, there is still a lot that is unknown about the sense of smell in this species and birds in general.

Response: We have rephrased this section to avoid the unintended strong expression of wording. 

Secondly, assuming house sparrows do have a great sense of smell and use olfaction to evaluate mates, there seems to be two possible explanations that could be driving the lack of significant findings. First, as I discussed above, the body odor of house sparrows may contain no information about MHC-I genes. I think this should be mentioned in the discussion

Response: Thank you for your comments. We have now added this as a potential explanation to the discussion (lines 482-488): “Similarly, further research is needed to elucidate the source of MHC-I-related scent cues in birds. One of the major outstanding questions is whether house sparrow body odour does indeed convey information about the genetic make-up of MHC-I. We therefore propose further experiments examining whether chemical profiles vary in relation to MHC-I characteristics of individuals and to determine whether these potential differences in MHC-I characteristics can be assessed through olfaction in birds.”

Moreover, another potential factor driving the results of this manuscript is the influence of other MHC genes or other genetic markers on the odor of the birds, and thus their preferences in this experiment. Most of the evidence in birds that connects chemical profiles, olfaction, and social interactions has focused on MHC-II. This has been the focus because of its connection to the microbiome and the ability of bacteria produce volatile chemicals. As is pointed out in this manuscript, birds lack a vomeronasal organ and therefore likely only sense volatile chemicals using their olfactory system. Volatile information about MHC-II may be more important to house sparrows and driving their choices in this experiment. Furthermore, non-MHC genes can shape chemical profiles, and birds could be using odors associated with other genetic markers to form preferences. I would like to see more discussion around how other genes that were not examined in this study could be driving the results.

Response: We have mentioned that “ Evidence that birds are using the volatile compound of uropygial gland secretions for detecting the MHC-II characteristics of conspecifics [29] comes from a study in song sparrows that were able to assess MHC-II similarity when exposed only to the uropygial gland secretion. While a recent body of evidence suggests that MHC-II genes play a critical role in mate choice that can be assessed by olfaction as shown for blue petrels [28] and song sparrows [29], whether house sparrows employ olfaction to assess MHC-II genes or other genetic characteristics of potential partners is still not known.”

Similarly, further research is needed to elucidate the source of MHC-I-related scent cues in birds. One of the major outstanding questions is whether house sparrow body odour does indeed convey information about the genetic make-up of MHC-I. We therefore propose further experiments examining whether chemical profiles vary in relation to MHC-I characteristics of individuals and to determine whether these potential differences in MHC-I characteristics can be assessed through olfaction in birds. .”

One aspect of this study that I thought was particularly strong was the use of live birds as the scent source. This is in contrast to the Leclaire et al. 2017 which used cotton swabs and bags as the odor source and Grieves et al. which used a sample of preen oil. While preen oil is thought to be the primary source of chemicals that make up avian body odor, there are differences between the compounds on bird feathers and the preen oil (see Mardon et al. 2011 J Avian Biology https://onlinelibrary.wiley.com/doi/abs/10.1111/j.1600-048X.2010.05113.x) Arguably, using a live bird is the most accurate approach because it ensures all the odorants are present, even if the sounds and movements produced by live birds may complicate things. The authors may want to consider including slightly more discussion on their decision to use live birds and why this is a strength of their study.

Response: Thank you very much for your comments and remarks. We have tried to improve the discussion regarding the methodology used.

---

## [Editor Report · Decision Letter 2]

27 Oct 2022

PONE-D-22-00952R2House sparrows do not exhibit a preference for the scent of potential partners with different MHC-I diversity and genetic distancesPLOS ONE

Dear Dr. Amo,

Thank you for submitting your manuscript to PLOS ONE. After careful consideration, we feel that it has merit but does not fully meet PLOS ONE’s publication criteria as it currently stands. Therefore, we invite you to submit a revised version of the manuscript that addresses the points raised during the review process.

We look forward to receiving your revised manuscript.

Kind regards,

Magdalena Ruiz-Rodriguez

Academic Editor

PLOS ONE

Journal Requirements:

Additional Editor Comments (if provided):

In the manuscript PONE-D-22-00952R2, authors have made a great effort to improve the quality and clarity of the manuscript. They answered in detail every concern of the reviewers, and performed the requested changes. I appreciate that finally they present the analysis of genetic distances as continuous and categorical variables.

This manuscript is ready to be published in Plos One, but first, I would like to suggest some minor changes:

L178: include some explanation after “and”. E.g. “… of our reads and previous research (40)).

L384: Remove the comma after ).

L430: I think that there is a mistake, given that the preference was for the left side of the chamber.

L434: Is there any reference to support that the laterality does not affect the olfactory preferences of birds?
---

## [Author Response · Author response to Decision Letter 2]

28 Oct 2022

In the manuscript PONE-D-22-00952R2, authors have made a great effort to improve the quality and clarity of the manuscript. They answered in detail every concern of the reviewers, and performed the requested changes. I appreciate that finally they present the analysis of genetic distances as continuous and categorical variables.

Response: We would like to thank you and the referees for your comments that have substantially improved our ms.

This manuscript is ready to be published in Plos One, but first, I would like to suggest some minor changes:

L178: include some explanation after “and”. E.g. “… of our reads and previous research (40)).

Response: Thanks, we have added “and previous research”. 

L384: Remove the comma after ).

Response: Thanks, we have removed the comma.

L430: I think that there is a mistake, given that the preference was for the left side of the chamber.

Response: Thank you for taking notice of the mistake. In fact, we have realized that the mistake was in the Result Section, as preference was for the right size of the chamber. We have corrected in lines 382-384: “However, we found a general side bias impacting the analysis of the bird’s choice with more birds choosing the right side of the chamber (Z = -0.83, P = 0.03, and Z = -2.23, P = 0.03, respectively).”.

L434: Is there any reference to support that the laterality does not affect the olfactory preferences of birds?

Response: To our knowledge, there is no previous evidence that brain laterality affects olfactory preferences of birds. We have rephrased the paragraph (lines 430-435): “We found a slight preference for the right side of the chamber that may be attributed to hemispherical asymmetry. House sparrows exhibited laterality in the brain towards the right, as previously reported in other sparrow species, such as American tree sparrows (Spizella arborea) [67]. Our results suggest that this laterality does not seem to affect olfactory preferences of focal birds. However, further studies are needed to explore olfactory laterality in avian species.”

---

## [Editor Report · Decision Letter 3]

24 Nov 2022

House sparrows do not exhibit a preference for the scent of potential partners with different MHC-I diversity and genetic distances

PONE-D-22-00952R3

Dear Dr. Amo,

We’re pleased to inform you that your manuscript has been judged scientifically suitable for publication and will be formally accepted for publication once it meets all outstanding technical requirements.

Kind regards,

Magdalena Ruiz-Rodriguez

Academic Editor

PLOS ONE
---

## [Editor Report · Acceptance letter]

13 Dec 2022

PONE-D-22-00952R3 

House sparrows do not exhibit a preference for the scent of potential partners with different MHC-I diversity and genetic distances 

Dear Dr. Amo:

I'm pleased to inform you that your manuscript has been deemed suitable for publication in PLOS ONE. Congratulations! Your manuscript is now with our production department. 

Kind regards, 

on behalf of

Dr. Magdalena Ruiz-Rodriguez 

Academic Editor

PLOS ONE